

# Field evaluation of low-cost particulate matter sensors in high and low concentration environments

Tongshu Zheng[1], Michael H. Bergin[1], Karoline K. Johnson[1], Sachchida N. Tripathi[2], Shilpa Shirodkar[2], Matthew S. Landis[3], Ronak Sutaria[4], David E. Carlson[1,5]

[1]Department of Civil and Environmental Engineering, Duke University, Durham, NC 27708, USA
[2]Department of Civil Engineering, Indian Institute of Technology Kanpur, Kanpur, Uttar Pradesh 208016, India
[3]US Environmental Protection Agency, Office of Research and Development, Research Triangle Park, NC 27711, USA
[4]Center for Urban Science and Engineering, Indian Institute of Technology Bombay, Mumbai, Maharashtra 400076, India
[5]Department of Biostatistics and Bioinformatics, Duke University, Durham, NC 27708, USA

*Correspondence to*: Tongshu Zheng (tongshu.zheng@duke.edu)

**Abstract.** Low-cost particulate matter (PM) sensors are promising tools for supplementing existing air quality monitoring networks. However, the performance of the new generation of low-cost PM sensors under field conditions is not well understood. In this study, we characterized the performance capabilities of a new low-cost PM sensor model (Plantower model PMS3003) for measuring $PM_{2.5}$ at 1 min, 1 h, 6 h, 12 h and 24 h integration times. We tested the PMS3003s in both low concentration suburban regions (Durham and Research Triangle Park (RTP), NC, US) with 1 h $PM_{2.5}$ (mean ± Std.Dev) of 9 ± 9 μg m$^{-3}$ and 10 ± 3 μg m$^{-3}$ respectively, and a high concentration urban location (Kanpur, India) with 1 h $PM_{2.5}$ of 36 ± 17 μg m$^{-3}$ and 116 ± 57 μg m$^{-3}$ during monsoon and post-monsoon seasons, respectively. In Durham and Kanpur, the sensors were compared to a research-grade instrument (environmental β-attenuation monitor (E-BAM)) to determine how these sensors perform across a range of $PM_{2.5}$ concentrations and meteorological factors (e.g., temperature and relative humidity (RH)). In RTP, the sensors were compared to three Federal Equivalent Methods (FEMs) including two Teledyne Model T640s and a ThermoScientific Model 5030 SHARP to demonstrate the importance of the type of reference monitor selected for sensor calibration. The decrease of 1 h mean errors of the calibrated sensors using univariate linear models from Durham (201%) to Kanpur monsoon (46%) and to post-monsoon (35%) season showed that PMS3003 performance generally improved as ambient $PM_{2.5}$ increased. The precision of reference instruments (T640: ±0.5 μg m$^{-3}$ for 1 h; SHARP: ±2 μg m$^{-3}$ for 24 h, better than the E-BAM) is critical in evaluating sensor performance and β-attenuation-based monitors may not be ideal for testing PM sensors at low concentrations, as underscored by 1) the less dramatic error reduction over averaging times in RTP against optical-based T640 (from 27% for 1 h to 9% for 24 h) than in Durham (from 201% to 15%); 2) the lower errors in RTP than Kanpur post-monsoon season (from 35% to 11%); 3) the higher T640–PMS3003s correlations ($R^2 \geq 0.63$) than SHARP–PMS3003s ($R^2 \geq 0.25$). A major RH influence was found in RTP (1 h RH = 64 ± 22%) due to the relatively high precision of the T640 measurements that can explain up to ~30% of the variance in 1 min to 6 h PMS3003 $PM_{2.5}$ measurements. When proper RH corrections are made by empirical non-linear equations after using a more precise reference method to calibrate the sensors, our work suggests that the PMS3003s can measure $PM_{2.5}$ concentrations within ~10% of ambient values. We observed that PMS3003s appeared to exhibit a non-linear response when ambient $PM_{2.5}$



exceeded ~125 µg m$^{-3}$ and found that the quadratic fit is more appropriate than the univariate linear model to capture this nonlinearity and can further reduce errors by up to 11%. Our results have substantial implications for how variability in ambient PM$_{2.5}$ concentrations, reference monitor types, and meteorological factors can affect PMS3003 performance characterization.

## 5  1 Introduction

Exposure to particulate matter (PM) is associated with cardiopulmonary morbidity and mortality. Multiple complex pathophysiological or mechanistic pathways have been identified as the underlying cause of this association (Pope and Dockery, 2006). Fine particles (PM$_{2.5}$, with a diameter of 2.5 µm and smaller) pose a greater threat to human health than their larger and coarser counterparts due to their higher levels of toxicity, stronger tendency towards deposition deep in the

lungs, and longer lifetime in the lungs (Pope and Dockery, 2006). From an environmental perspective, PM$_{2.5}$ contributes to decreased visibility, environmental damages such as depletion of soil nutrients, acid rain effects, and material damages such as discoloration of the Taj Mahal (US Environmental Protection Agency (US EPA), 2016a; Bergin et al., 2015).

In the US, PM$_{2.5}$ is regulated and monitored under the National Ambient Air Quality Standards (NAAQS) (US EPA, 2016b).

The NAAQS compliance monitoring approves the use of both the Federal Reference Methods (FRMs) and the Federal Equivalent Methods (FEMs) to accurately and reliably measure PM$_{2.5}$ in outdoor air (US EPA, 2017). While these kinds of instruments provide measurements of decision-making quality, they require skilled staff, close oversight, regular maintenance, and stringent environmental operating conditions (Chow, 1995). The personnel, infrastructure, and financial demands of running a regulatory PM$_{2.5}$ monitor make it impractical to deploy them in a dense monitoring network and make

it consequently hard to gather high temporally and spatially resolved air quality information. The lack of fine-grained PM$_{2.5}$ monitoring data hinders the characterization of urban PM$_{2.5}$ gradients/distributions (Kelly et al., 2017), and prohibits exposure scientists from adequately quantifying the relationship between air pollution exposures and health effects (Holstius et al., 2014). The lack of finely resolved ambient PM$_{2.5}$ data also restricts prompt empirical verifications of emission-reduction policies and inhibits rapid screening for urban "hot spots" (Holstius et al., 2014).

These conventional techniques' deficiencies in measuring PM$_{2.5}$ along with the technological advancements in multiple areas of electrical engineering (Snyder et al., 2013) foster a paradigm shift to the use of small, portable, inexpensive, and real-time sensor packages for air quality measurement. As these sensors can provide almost instantaneous feedback about changes in air quality and at a low cost, citizens may be more willing to be part of "participatory measurement" including determining if

they are in areas with high levels of pollution and exploring how to decrease their exposure. Air pollution control agencies such as the South Coast Air Quality Management District (SCAQMD) have already been researching ways of empowering



local communities to answer questions about their specific air quality issues with sensors and potentially engaging them in future projects (US EPA, 2016c).

Previous evaluations of numerous low-cost PM sensor models have demonstrated promising results in comparison with FEMs or research-grade instruments in some field studies. These models include Shinyei PPD20V (Johnson et al., 2018), Shinyei PPD42NS (Holstius et al., 2014; Gao et al., 2015), Shinyei PPD60PV (SCAQMD, 2015a; Jiao et al., 2016; Mukherjee et al., 2017; Johnson et al., 2018), AlphaSense OPC-N2 (SCAQMD, 2015b; Mukherjee et al., 2017; Crilley et al., 2018), Plantower PMS1003 (Kelly et al., 2017; SCAQMD, 2017b), Plantower PMS3003 (SCAQMD, 2017a), and Plantower PMS5003 (SCAQMD, 2017c). Currently, all Plantower PMS models have only been tested at low to moderately high

ambient $PM_{2.5}$ concentrations in US. Kelly et al. (2017) assessed the performance of Plantower PMS1003 against an FRM, two FEMs, and a research-grade instrument in a 41-day field campaign in the southeast region of Salt Lake City during winter. They reported both high 1 h PMS–FEMs $PM_{2.5}$ correlations ($R^2$ = 0.83–0.92) and high 24 h PMS–FRM $PM_{2.5}$ correlations ($R^2$ > 0.88). The SCAQMD's Air Quality Sensor Performance Evaluation Center (AQ-SPEC) has field-tested Laser Egg Sensor (Plantower PMS3003 sensors), PurpleAir (Plantower PMS1003 sensors), and PurpleAir PA-II (Plantower

PMS5003 sensors) with triplicates per model located next to FEMs at ambient monitoring sites in Southern California for a roughly 2-month period (SCAQMD, 2017a, 2017b, 2017c). Even though the evaluation results are still preliminary, they filled in gaps in the documentation of the performance of the new generation of low-cost PM sensors. The SCAQMD found that both PMS1003 and PMS5003 raw $PM_{2.5}$ measurements correlated very well with the corresponding FEM GRIMM Model 180 ($R^2$ > 0.90 and $R^2$ > 0.93, respectively) and FEM BAM-1020 ($R^2$ > 0.78 and $R^2$ > 0.86, respectively). The

SCAQMD, however, reported a moderate correlation between 1 h raw PMS3003 $PM_{2.5}$ measurements and the corresponding FEM BAM-1020 ($R^2$ ~ 0.58).

Despite the favorable correlation of these sensors in comparison with reference monitors during these field evaluations, considerable challenges have also been acknowledged. To date, there is only limited understanding of the performance

specifications of these emerging low-cost PM sensor models (Lewis and Edwards, 2016). This situation is further confounded by the fact that a model's agreement with reference instruments, and the corresponding calibration curves established vary with the operating conditions (RH, temperature, and $PM_{2.5}$ mass concentrations), the aerosol properties (aerosol composition, size distribution, and the resulting light scattering efficiency), and the choice of reference instruments (Holstius et al., 2014; Gao et al., 2015; Kelly et al., 2017). Artifacts such as varying RH and temperature significantly

interfere with accurate reporting of $PM_{2.5}$ results from low-cost PM sensors. To the best of our knowledge, only Crilley et al. (2018) have adequately compensated for the RH bias in low-cost PM sensor measurements based on $\kappa$-Köhler theory and they found a roughly 1 order of magnitude improvement in the accuracy of sensor measurements after correcting for RH bias. Also, US EPA FEMs are required to provide results comparable to the FRMs only for a 24 h but not a 1 h sampling



period. An inappropriate selection of reference monitors in field tests (especially in low $PM_{2.5}$ concentration environments) might prejudice the overall performance of low-cost sensors' short-term measurements.

These limitations in the previous scientific work warrant more testing under diverse ambient environmental conditions alongside various reference monitors, and more rigorous methods (statistical and calibration) to characterize a particular low-cost sensor model's performance. It is of paramount importance to quantify the accuracy and precision of these sensors, as the value of the rest of the related work such as data analyses, sensor network establishment, and citizen engagement is conditional on this. This paper focuses on 1) comparing a new low-cost PM sensor model (Plantower PMS3003) to different reference monitors (including a newly designated US EPA $PM_{2.5}$ FEM, i.e., Teledyne API T640 PM mass monitor) in both high (Kanpur, Uttar Pradesh, India 1 h $PM_{2.5}$ average $\geq$ 36 µg m$^{-3}$) and relatively low (Durham and Research Triangle Park, NC, US 1 h $PM_{2.5}$ average $\leq$ 10 µg m$^{-3}$) ambient $PM_{2.5}$ concentration environments; 2) calculating metrics including mean of ratios and error in addition to correlation coefficient ($R^2$) to more rigorously interpret low-cost sensors' performance capabilities as a function of averaging timescales; 3) conducting appropriate RH and temperature adjustments when possible to sensor $PM_{2.5}$ responses in order to account for systematic meteorology-induced influences and consequently to present $PM_{2.5}$ measurements with relatively high accuracy and precision at a low cost. To our knowledge, this is the first study to evaluate such a low-cost PM sensor model under high ambient conditions during two typical and distinct seasons (i.e., monsoon and post-monsoon) in India, and the first to use the T640 PM mass monitor (Teledyne API) as a reference monitor to examine sensor performance.

## 2 Materials and methods

### 2.1 Sensor configuration

The low-cost sensors evaluated in the present study are Plantower particulate matter sensors (model PMS3003). The Plantower PMS3003 sensors were chosen because 1) they are priced at a small fraction of the cost of reference monitors (approximately USD 30) and 2) their manufacturer reported maximum errors are relatively low (±10 µg m$^{-3}$ in the 0–100 µg m$^{-3}$ range, and ±10% in the 100–500 µg m$^{-3}$ range). The sensors employ a light-scattering approach to measure $PM_1$, $PM_{2.5}$, and $PM_{10}$ mass concentrations in real-time. Ambient air laden with different-sized particles is drawn into the sensor measurement volume where the particles are illuminated with a laser beam, and the resulting scattered light is measured perpendicularly by a recipient photo-diode detector. These raw light signals are filtered and amplified via electronic filters and circuitry before being converted to mass concentrations. The manufacturer datasheet indicates that the measurement range of this specific sensor model spans from 0.3 µm to 10 µm. PM mass concentration measurements either with or without a manufacturer "atmospheric" calibration are available from the Plantower sensor outputs. Nevertheless, the manufacturer did not provide any documentation to elaborate on how the calibration algorithm was derived. The influence of





meteorological factors (e.g., RH, temperature) was likely not accounted for in the manufacturer calibrations. Therefore, we used the sensor reported PM concentration estimates without an "atmospheric" calibration in the current study. Prior to field deployment, no attempt was made to calibrate these sensors under laboratory conditions due to a potentially marked discrepancy in particle size, composition, and optical properties of field and laboratory conditions.

The Plantower PMS3003 sensor along with a Sparkfun SHT15 RH and temperature sensor, a Teensy 3.2 USB-based microcontroller, a ChronoDot V2.1 high precision real-time clock, a microSD card adapter, a Pololu 5V S7V7F5 voltage regulator, a DC barrel jack connector, and a basic 5 mm LED was connected to a custom designed printed circuit board (PCB), shown in Fig. 1a. We programmed the Teensy 3.2 microcontroller to measure PM mass concentrations ($\mu$g m$^{-3}$)

every second and to store the time-stamped 1 min averaged measurements to text files on a microSD card. To protect sensors from rain and direct sunlight, all components were housed in a 20.50 cm L × 9.95 cm W × 6.70 cm H NEMA (National Electrical Manufacturers Association) electrical box (Bud Industries NBF32306) as shown in Fig. 1b. The inlet of the Plantower sensor was aligned with a hole drilled in the electrical box to ensure unrestricted airflow into the sensor. The Duke PM air quality monitoring packages were continuously powered up by 5V 1A USB wall chargers. The total material costs

for one PM monitoring package including the Plantower PMS3003 sensor, the supporting circuitry, the enclosure, and additional power cords are approximately USD 200. More detailed instructions on how to assemble the sensor packages and information on how to use their data can be found on our webpage (http://dukearc.com).

## 2.2 Field deployment

Three field campaigns were conducted to evaluate the performance characteristics of Plantower PMS3003 sensors and to

explore the potential impacts from artifacts such as RH and temperature on sensors' PM$_{2.5}$ measurements (Table 1). Two sites were in Durham County, NC, representing suburban environments with low ambient PM$_{2.5}$ concentrations. The other study site was in Kanpur, Uttar Pradesh, India, representing an urban-influenced environment. The data from Kanpur were subset into the monsoon season with moderately high PM$_{2.5}$ concentrations, and the post-monsoon season with high PM$_{2.5}$ concentrations.

**2.2.1 Low concentration region: Durham and Research Triangle Park (RTP), NC**

The first measurement campaign in the low concentration region was on the rooftop of the Fitzpatrick Center, a three-story building located on the Duke University West Campus in Durham, NC (Latitude: 36.003350, Longitude: -78.940259). The sampling location lies in close proximity to the 7,052-acre Duke Forest and approximately 3.5 km from the Durham downtown and 4.5 km from the Durham National Guard Armory monitoring station (Latitude: 36.0330, Longitude: -

78.9043). This study location is also about 950 m southwest of the Durham Freeway, which had an annual average daily traffic of 43,000 vehicles as of 2015 (North Carolina Department of Transportation, 2015). No known principle point source emissions are located in the surrounding area. The 3-year average (2013–2015) for PM$_{2.5}$ concentrations reported by the



Durham National Guard Armory monitoring station was 12 µg m$^{-3}$, and the reported 98$^{th}$ percentile daily average from 2013 to 2015 was 18 µg m$^{-3}$ (North Carolina Department of Environmental Quality, 2017). At the Duke site, five Plantower PMS3003 sensors (labeled PMS3003-1 through -5) were compared to a collocated Environmental β-Attenuation Monitor E-BAM-9800 (Met One Instruments). Unlike its more advanced counterpart BAM-1020 (Met One Instruments), the E-BAM-

9800 is not currently a US EPA designated FEM for PM$_{2.5}$ mass concentration continuous monitoring, although it is ideal for rapid deployment because of its portability and its ability to accurately track FRM or FEM results with proper operation and regular maintenance (Met One Instruments, 2008). The hourly values reported by the E-BAM (in mg m$^{-3}$) were used in the analyses. The E-BAM's sporadic negative values caused by low actual ambient concentrations (such as below 3 µg m$^{-3}$) were replaced with 0 µg m$^{-3}$ in this study. The sensor packages were strapped to the E-BAM tripod and operated in a

collocated manner for a period of 50 days from February 1, 2017 to March 31, 2017 (all the sensor packages and the E-BAM were shut down between March 3 and March 12 for maintenance). Over the course of the deployment, PMS3003-1 was disconnected between February 14 and February 21 because of power supply issues, and this situation rendered PMS3003-1 data 86% complete.

The second ambient test in the low concentration region was performed at the US EPA's Ambient Air Innovation Research Site (AIRS) on its RTP campus, NC (Latitude: 35.882816, Longitude: -78.874471) about 16 km southeast of the Duke site. The ambient PM$_{2.5}$ mass concentrations in the RTP region are normally well under 12 µg m$^{-3}$ (Williams et al., 2003). A Thermo Scientific 5030 SHARP (Synchronized Hybrid Ambient Real-time Particulate Monitor) monitor (US EPA PM$_{2.5}$ FEM) was operated by the US EPA Office of Research and Development (ORD) and two Teledyne API T640 PM mass

monitors (US EPA PM$_{2.5}$ FEM) were operated by the US EPA Office of Air Quality Planning and Standards (OAQPS). The SHARP monitor is a hybrid of a high sensitivity nephelometer using 880 nm Infrared Light Emitting Diodes (IREDs) and a BAM. The SHARP continuously calculates the ratios of dynamically time-averaged beta concentrations to dynamically time-averaged nephelometer concentrations, and continuously employs these ratios as correction factors to adjust the raw 1 min averaged nephelometer readings. The corrected nephelometric concentrations are reported as 1 min SHARP

measurements in µg m$^{-3}$ (Thermo Fisher Scientific, 2007). The T640 monitor, first introduced in 2016, is one of the latest additions to the list of approved US EPA PM$_{2.5}$ FEM monitors. The T640 is essentially an optical aerosol spectrometer that uses light scattering to measure particle diameters in 256 particle size classes over 0.18–20 µm range at the single particle level. The 256 size classes are subsequently combined into 64 channels for mass calculation with proprietary algorithms. The light source used by the T640 monitor is polychromatic (broadband) light. Compared to traditional monochromatic laser

scattering approaches, the polychromatic light approach provides more robust and accurate measurements with significantly less noise especially over the particle size range of 1 µm to 10 µm (Teledyne Advanced Pollution Instrumentation, 2016). The T640 reports 1 min resolution results in µg m$^{-3}$. The SHARP and one of the T640 units (T640_Shelter) were installed inside an ORD mobile laboratory and an OAQPS shelter, respectively with roof penetration while the other T640 unit



(T640_Roof) was installed inside an outdoor enclosure with heating, ventilation and air conditioning (HVAC) control on the rooftop of the OAQPS shelter. Three PMS3003 sensor packages from the Duke site (labeled PMS3003-1 through -3) were attached to the rail on top of the ORD mobile laboratory approximately 3 to 4 m above ground. The SHARP inlet and the sensor packages' inlets were only a few feet apart. The two T640 inlets were situated on the rooftop of the OAQPS shelter,

within about 30 m of the sensor packages' inlets. The inlets of these instruments were positioned roughly at the same height above ground. Over the course of the 32-day field project (June 30, 2017 to July 31, 2017), all the instruments' data completeness was 100% except the SHARP (99%). The slightly incomplete SHARP data stemmed from the removal of midnight concentration spikes (at approximately 01:00 to 01:10 am) due to the daily filter tape advancement.

### 2.2.2 High concentration location: Indian Institute of Technology Kanpur (IIT Kanpur) study site

Identical to the set-up at the Duke site, the third field evaluation involving two PMS3003 sensors (labeled PMS3003-6 and -7) alongside an E-BAM was carried out on the rooftop of the Center for Environmental Science and Engineering inside the campus of IIT Kanpur (Latitude: 26.515818, Longitude: 80.234337). The Center is a two-story building (roughly 12 m above the ground level) that lies approximately 15 km northwest of downtown Kanpur city. The institute is located upwind of Kanpur city and away from major roadways, industrial sites, and dense residential communities, therefore it has

comparatively low $PM_{2.5}$ concentrations (Villalobos et al., 2015). Kanpur is a heavily polluted industrial city on the Indo-Gangetic Plain with a large urban area of dense population (approximately 2.7 million) (Villalobos et al., 2015). Various small-scale industries, a coal-fired power plant (Panki Thermal Power Station), indoor and outdoor biomass burning, heavy vehicles on the Grand Trunk Road (a major national highway) running through Kanpur city, fertilizer plants, and refineries are the prime contributors to air pollution (Shamjad et al., 2015; Villalobos et al., 2015). The local climate is primarily

defined as humid subtropical with extremely hot summers and cold winters (Ghosh et al., 2014). The monsoon season (June to September) is documented to have lower $PM_{2.5}$ concentrations than the post-monsoon season (October and November) (Bran and Srivastava, 2017). The two sensor packages were first deployed at the study site on June 8, 2017 for approximately 22 days (early monsoon), and then on October 23, 2017 for approximately 25 days (post-monsoon). Since these two sensor units were not embedded with temperature and RH sensors, the temperature and RH data (available as 15

min averages) were simultaneously collected from an automatic weather station, roughly 500 m away from the study site and 2 to 3 m above ground. Throughout the sampling periods, error-flagged E-BAM measurements (including delta temperature setpoint exceeded, flow failure, abnormal flow rate, beta count failure) during the operation were excluded from the analyses for quality assurance purposes, and this caused the E-BAM data to be 85% and 93% complete for monsoon and post-monsoon seasons, respectively. The two sensor packages had data completeness close to 100% for both monsoon and post-

monsoon seasons. The temperature and RH data from the automatic weather station were only occasionally missing due to power supply issues with an overall 93% and 99% completeness for monsoon and post-monsoon seasons, respectively.



### 2.3 Sensor calibrations

Sensor PM$_{2.5}$ measurement adjustments/corrections were made as described in the following three subsections. First, we evaluate the dependence of sensor response on RH (Sect. 2.3.1), if this was significant we adjusted sensor PM$_{2.5}$ values for RH. Next, we investigated the sensor response dependency on temperature (Sect. 2.3.2), if this was significant we

simultaneously adjusted sensor PM$_{2.5}$ values for temperature and calibrated sensor values based on reference monitors. If this was not significant, we simply applied a calibration based on the reference PM$_{2.5}$ values and corrected for any non-linear performance (Sect. 2.3.3). The calibration strategy is shown graphically in Fig. 2.

### 2.3.1 RH adjustment to sensor PM$_{2.5}$ measurements

FEMs and research-grade PM analyzers typically control for RH by dynamically heating the sample air inlet. Our sensor

packages, similar to many low-cost designs, are not equipped with any heaters/conditioners to reduce RH impact. Therefore, the RH can significantly bias the PM$_{2.5}$ mass concentrations reported by our sensor packages. The effect of RH on the mass of atmospheric aerosol particles has been well-documented for decades. Sinclair et al. (1974) showed that there was a 2 to 6-fold increase in the mass of particles, depending on the properties of the particles, as the RH reached 100%. Waggoner et al. (1981) also showed that RH above roughly 70% can enhance scattering coefficients of hygroscopic or deliquescent particles

in various locations in the west and mid-west US due to the growth of these particles associated with water uptake. Zhang et al. (1994) described the calculated scattering efficiencies of ammonium sulfate in the Grand Canyon as a function of RH with empirical Eq. (1). This equation was later employed by Chakrabarti et al. (2004) to predict the effect of RH on the relationship between the nephelometric personal monitors' PM$_{2.5}$ mass concentration measurements and the results of a reference monitor (BAM). They found that the model fitted field data collected from their study and a previous study (Day

and Malm, 2000) quite well. An identical equation was also among a wide variety of approaches assessed by Soneja et al. (2014) to adjust nephelometric personal monitor PM$_{2.5}$ readings for the RH impact. We believe lessons learned from these previous studies can be directly applied to RH adjustments for low-cost nephelometric sensors' PM$_{2.5}$ measurements in the present study by using Eq. (1):

$$\text{RH correction factor} = \frac{\text{scattering efficiency (for a given RH)}}{\text{scattering efficiency (RH=30\%)}} = \frac{\text{raw PMS3003 PM}_{2.5} \text{ conc. (for a given RH)}}{\text{reference PM}_{2.5} \text{ conc. (for a given RH)}} = a + b \times \frac{\text{RH}^2}{1\text{-RH}} \qquad (1)$$

Ordinary least squares (OLS) regressions were conducted to obtain the empirical regression parameters $a$ and $b$ in Eq. (1), where the dependent variable was RH correction factors calculated as the ratio of PMS3003 PM$_{2.5}$ mass concentrations averaged across all the sensor package units to the corresponding reference monitor concentrations at each point in time at a sampling location, and the independent variable was the entire RH$^2$ / (1 - RH) term. The RH was the measurements averaged

across all the embedded Sparkfun SHT15 RH and temperature sensors at each point in time for the calibration models of Duke University and EPA RTP study sites, and the measurements from the automatic weather station for the models of IIT Kanpur study site. The empirical equations derived were used to compute RH correction factor for a given RH at the



sampling sites. The RH interferences were compensated by dividing each individual raw PMS3003 PM$_{2.5}$ mass concentration for a given RH by the RH correction factor yielded for that RH (Eq. (2)):

$$\text{RH adjusted PMS3003 PM}_{2.5} \text{ conc.} = \frac{\text{raw PMS3003 PM}_{2.5} \text{ conc. (for a given RH)}}{\text{RH correction factor (for a given RH)}} \qquad (2)$$

We only performed the RH adjustments when the fitted models for any of the sampling locations over any time averaging interval had at least a moderate coefficient of determination ($R^2 \geq 0.40$). Despite the similarity of the general shape of correction factor curves in different studies, the detailed behaviors of aerosols diverged greatly due to considerable difference in particles' chemical composition and diameter (Waggoner et al., 1981; Zhang et al., 1994; Day and Malm, 2000; Chakrabarti et al., 2004; Soneja et al., 2014). In a previous study (Day and Malm, 2000), aerosols mass at some locations

began to increase continuously above a relatively low RH (such as 20%), whereas at other locations it exhibited a distinct deliquescent behavior (i.e., aerosols water uptake occurred at a relatively high RH). Even for aerosols showing deliquescent behavior, the observed deliquescence RH (RH threshold) varies from study to study. Soneja et al. (2014) also found underestimation of PM concentrations (correction factors less than 1) below 40% RH. Because of these uncertainties, we conducted RH adjustments across the entire range of recorded RH without incorporating an RH threshold. Additionally, the

RH adjustments in this study were always performed separately from and prior to either temperature adjustments or reference monitor adjustments.

### 2.3.2 Temperature adjustment to sensor PM$_{2.5}$ measurements

The Akaike's Information Criterion (AIC) is a widely used tool for model selection to address the fact that including additional predictors may overfit the data (Crawley, 2017a). It was used to determine the significance of the temperature

term in the PMS3003 calibration models for all the study locations at various averaging times. The AIC penalizes more complex models based on the number of parameters fit in that model. A lower AIC when comparing two models for the same data set indicates a better fitting model. In a linear regression model, an AIC difference between two models of less than or equal to 2 indicates that the more complex model does not improve predictive performance. Therefore, the simpler model should be adopted. We specifically compared the AIC value of a multiple linear regression model, which included

both the reference monitor measurement and temperature as predictor variables and without considering an interaction term (i.e., Eq. (3)) to the value of a univariate linear regression model with only the reference monitor measurement as a predictor variable (i.e., Eq. (4)). We performed the temperature adjustments using Eq. (5) only when the AIC indicated that the temperature predictor was significant in the calibration model (i.e., AIC $_{\text{Eq. (4)}}$ - AIC $_{\text{Eq. (3)}}$ > 2).

$$[\text{raw (or RH adjusted) PMS3003 PM}_{2.5} \text{ conc.}] = \beta_0 + \beta_1 \times \text{reference PM}_{2.5} \text{ conc.} + \beta_2 \times \text{temperature} \qquad (3)$$

$$[\text{raw (or RH adjusted) PMS3003 PM}_{2.5} \text{ conc.}] = \beta_0 + \beta_1 \times \text{reference PM}_{2.5} \text{ conc.} \qquad (4)$$



[temperature and reference monitor (and RH) adjusted PMS3003 PM$_{2.5}$ conc.] $=$

$$\frac{\text{[raw (or RH adjusted) PMS3003 PM}_{2.5} \text{ conc.]} - \beta_0 - \beta_2 \times \text{temperature}}{\beta_1} \tag{5}$$

The temperature was the measurements averaged across all the embedded Sparkfun SHT15 RH and temperature sensors at
each point in time for the models of Duke University and EPA RTP study sites, and the measurements from the automatic
weather station for the models of IIT Kanpur study site. Since the RH adjustments in this study were always performed first,
the PMS3003 PM$_{2.5}$ conc. in Eq. (3) and Eq. (4) were RH adjusted PMS3003 PM concentrations when RH adjustments were
significant, and were otherwise raw PMS3003 PM$_{2.5}$ concentrations. Additionally, temperature adjustments and reference
monitor adjustments were always conducted simultaneously when the temperature predictor was significant because Eq. (3)
consists of both the reference monitor concentration and temperature terms as independent variables. The AIC values for
models with 24 h data are not reported in the present study as 24 h observations generally have limited statistical power to
determine the significance of temperature in the models.

### 2.3.3 PM$_{2.5}$ sensor calibrations based on reference monitor values

The most basic calibration is a direct comparison with reference monitor measurements. We derived reference instrument
calibration equations (Eq. (4)) by fitting a linear least squares regression model to each pair of PMS3003 (dependent
variable) and collocated reference instrument's PM$_{2.5}$ mass concentrations (independent variable). The PMS3003 PM$_{2.5}$
values were RH adjusted concentrations when RH adjustments were significant and were otherwise raw concentrations. Each
PMS3003 measurement was subsequently calibrated using Eq. (6).

When the relationship between PM$_{2.5}$ mass concentrations of reference monitors and PMS3003 sensors was non-linear, PM$_{2.5}$
sensor calibration equations based on reference monitor values in a quadratic form (Eq. (7)) were used to describe the non-
linear performance and each PMS3003 measurement was subsequently calibrated using Eq. (8) since calibrated values
should always be on the left side of the axis of symmetry of the parabola with $a_2 < 0$. The AIC values (discussed in Sect.
2.3.2), and the root mean square errors (RMSE) (Eq. (9)) were used in combination to assess the goodness of fit and
accuracy of the two model approaches (i.e., univariate linear and quadratic models) as a function of integration times.

[reference monitor (and RH) adjusted PMS3003 PM$_{2.5}$ conc.] $= \dfrac{\text{[raw (or RH adjusted) PMS3003 PM}_{2.5} \text{ conc.]} - \beta_0}{\beta_1}$  (6)

[raw (or RH adjusted) PMS3003 PM$_{2.5}$ conc.] $= a_0 + a_1 \times$ reference PM$_{2.5}$ conc. $+ a_2 \times$ (reference PM$_{2.5}$ conc.)$^2$ (7)

[reference monitor (and RH) adjusted PMS3003 PM$_{2.5}$ conc.] $=$

$$\frac{-a_1 + \sqrt{a_1{}^2 - 4a_2 \times (a_0 - \text{[raw (or RH adjusted) PMS3003 PM}_{2.5} \text{ conc.]})}}{2a_2} \tag{8}$$



$$RMSE = \sqrt{\frac{1}{n}\sum_{i=1}^{n}(\hat{y}_i - y_i)^2} \tag{9}$$

where n is the number of observations, $\hat{y}_i$ is the calibrated PMS3003 PM$_{2.5}$ mass concentrations, and $y_i$ is the raw PMS3003 PM$_{2.5}$ mass concentrations.

**2.4 Sensor performance metrics**

Metrics such as the intercept, slope, and coefficient of determination ($R^2$) obtained from OLS models of sensor outputs with reference instrument measurements are widely used to evaluate sensor performance (Holstius et al., 2014; Gao et al., 2015; Wang et al., 2015; Jiao et al., 2016; Cross et al., 2017; Kelly et al., 2017; Zimmerman et al., 2018). In this study, all the $R^2$ in figures represent regression coefficients of the (calibration) equations while all the $R^2$ in tables represent regression

coefficients between the calibrated sensor and reference measurements. To date, only a few studies have attempted to measure parameters other than $R^2$ to gauge the overall performance of low-cost sensor technologies. They typically focus on the RMSE (Holstius et al., 2014; Cross et al., 2017; Zimmerman et al., 2018), the mean absolute error (MAE) and the mean bias error (MBE) (Cross et al., 2017; Zimmerman et al., 2018), and normalized residuals (Sousan et al., 2017; Kelly et al., 2017). In addition to the intercept, slope, and $R^2$, we also used ratios of the calibrated PMS3003 PM$_{2.5}$ mass concentrations

to reference monitor values to examine sensors' post-calibration performance. From this set of ratios, we calculated an average ratio and 1 standard deviation (Std.Dev), which are defined as mean of ratios and error for each sensor unit, respectively. The mean of ratios should be close to 1 after calibration, and we would expect the error of any PM$_{2.5}$ mass concentration reported by a particular PMS3003 unit to be within ± 1 Std.Dev × 100% for 68% of the time. Knowing the performance of calibrated PMS3003 sensors is particularly important for understanding these sensors' potential for future

applications such as investigating the source and transport patterns of PM in an urban environment or examining the effectiveness of certain PM abatement strategies.

While longer averaging times (i.e., ≥ 24 hours) typically smooth out noisy signals and result in enhanced sensors performance, shorter averaging times (i.e., hours or minutes) are of growing interest particularly in the field of exposure

assessment (Williams et al., 2017). Similar to Williams et al. (2017), we also evaluated sensor performance over a wide range of time averaging intervals, namely 1 min (for the EPA RTP – the only site where 1 min reference data were available), 1 h, 6 h, 12 h, and 24 h. The purpose of such an examination is to better understand the trade-off between errors and averaging times when using this type of sensor so that data accuracy and precision can be weighed against the need for highly time-resolved data for various desirable research or citizen science applications.



# 3 Results and discussion

## 3.1 Duke University rooftop low ambient PM₂.₅ concentration environment with E-BAM as the reference monitor

### 3.1.1 PM₂.₅ concentration, RH, and temperature on 1 h scale

Table 1 shows the summary statistics for 1 h averaged measurements at Duke University from February 1, 2017 to March 31, 2017. The 1 h E-BAM PM$_{2.5}$ measurements averaged $9 \pm 9 \mu g\ m^{-3}$. The hourly PM$_{2.5}$ averages of the uncalibrated sensors were close to that of the E-BAM and had little intra-sensor variability. We calculated the coefficient of variation (defined as the ratio of the Std.Dev and the mean of the PM$_{2.5}$ readings from the five replicate PMS3003 sensors) as an indicator of sensor precision which yielded 10%, indicating the relatively high precision of the PMS3003 model. RH and temperature averaged $45 \pm 19\%$ and $15 \pm 8°C$, respectively. Figure 3 compares the 1 h E-BAM PM$_{2.5}$ mass concentrations to the results of the five uncalibrated sensors. Overall, the uncalibrated PMS3003 measurements followed the trend in ambient PM$_{2.5}$ concentrations and were very responsive to most sudden spikes in concentrations. However, the sensors tended not to track the E-BAM well below $\sim 10\ \mu g\ m^{-3}$.

### 3.1.2 PMS3003 performance characteristics on various timescales

Correlations among the five uncalibrated PMS3003 units were high ($R^2 = 0.98–1.00$) even under low ambient PM$_{2.5}$ concentrations with slopes averaging $1 \pm 0.1$ and negligible intercepts averaging $0.3 \pm 0.3$ (Fig. S1), suggesting excellent intra-PMS3003 precision. Regressions of the uncalibrated 1 h and 24 h PM$_{2.5}$ measurements from the five PMS3003 units versus the corresponding E-BAM PM$_{2.5}$ values indicate that different PMS3003 sensor units generally had similar calibration factors (i.e., intercept and slope values) on the same timescale (Fig. 4). Comparing across the time averaging interval spectrum (Table 2), the calibration factors on different timescales were consistent with the exception of 1 h results. Raw 1 h aggregated PMS3003 PM$_{2.5}$ concentration measurements correlated only moderately with the corresponding E-BAM data with a mean $R^2$ of 0.40 (range: 0.36–0.41). When the averaging time increased from 1 h to 6 h, the $R^2$ showed a marked improvement (mean: 0.80, range: 0.77–0.82). When the averaging time further rose to 12 h and from 12 h to 24 h, although still accompanied by improvements in $R^2$ (mean: 0.84 and 0.93, respectively), the magnitudes of the improvements were considerably smaller than the one seen from 1 h to 6 h.

The SCAQMD (2017a) also field-tested three Plantower PMS3003 units (Laser Egg sensors) alongside an FEM (BAM-1020, Met One Instruments) over a study period of similar length (roughly 2 months) with similar ambient PM$_{2.5}$ concentrations (1 h PM$_{2.5}$ range: 0–40 μg m$^{-3}$) in Riverside, CA, although the data were presented differently (with reference and sensor measurements on y- and x-axis, respectively) and thus the values of calibration factors cannot be directly compared to our study. The SCAQMD study demonstrated the calibration factors on 1 h scale (intercept: 5.9–6.3, slope: 0.50–0.57) were virtually the same as the values on 24 h scale (intercept: 6.0–6.3, slope: 0.48–0.57). This observation is in contrast to our finding where 1 h results (intercept: 3.2–4.1, slope: 0.64–0.79) differed dramatically from the 24 h values



(intercept: -4.6–3.6, slope: 1.5–1.8). This discrepancy might stem from the use of different reference instruments in the two studies. While both instruments use beta attenuation as the measurement principle, the accuracy of BAM-1020 (FEM) for 1 h measurements in the SCAQMD study is significantly better than that of the E-BAM-9800 (research-grade) in our study. This may also account for the higher $R^2$ on 1 h scale in the SCAQMD study (around 0.58).

Table 2 shows that the pattern of errors was aligned with our expectation, with each of the four time integration values having successively more accurate post-calibration PMS3003 $PM_{2.5}$ concentrations than all the previous time integration values (i.e., the error decreased as the averaging time increased). Furthermore, the steep gradient at which the mean error reduced over averaging time (from 201% for 1 h to 15% for 24 h) was unusual and most likely caused by E-BAM's poor signal-to-noise ratio in low concentrations with short real-time average periods. This finding points out that the precision of reference monitors is a critical factor in sensor evaluation, as discussed in detail in Sect. 3.2.2. It should be noted that the strong correlation on 6 h scale ($R^2$ mean = 0.8) did not translate into a low error (mean: 53%). This observation emphasizes the downside of overreliance on the correlation in the examination of sensor performance.

Figure S2 displays the relationship between PMS3003-to-E-BAM $PM_{2.5}$ ratio and RH on 1 h scale at Duke University. There was no apparent pattern of fractional increase in $PM_{2.5}$ weight measured by uncalibrated PMS3003 sensors with RH. Fitting the empirical RH correction factor model (i.e., Eq. (1) in Sect. 2.3.1) to these field data resulted in an $R^2$ close to 0. Examination of patterns and model fitting at longer averaging time intervals (i.e., 6 h, 12 h, and 24 h) yielded comparable results (not shown). These findings are indicative of the negligible impact of RH on PMS3003 $PM_{2.5}$ responses at Duke University. This lack of RH interference is believed to stem from a combination of infrequently high RH conditions during the winter months (only 12.5% and 4.0% of the entire time greater than 70% and 80%, respectively) and large measurement error inherent in the E-BAM under low $PM_{2.5}$ concentrations.

Table S1 demonstrates that the AIC differences between the calibration models with only a true $PM_{2.5}$ concentration term and the models incorporating an additional temperature term were greater than 2 for only the 1 h aggregated data, implying the calibration model with an added temperature term was significantly better than its simpler counterpart only on the 1 h scale. Therefore, the temperature adjustment was performed only for 1 h averaged PMS3003 responses at the Duke University study site. Counterintuitively, Table 2 shows that the temperature correction worsened the sensor performance by bringing the mean of ratios down from 0.97 to 0.90, and by bringing the error up from 201% to 207%. The deterioration in performance was likely to arise from large measurement error inherent in the E-BAM under low $PM_{2.5}$ concentrations.

**3.2 RTP low ambient $PM_{2.5}$ concentration environment with SHARP and T640 as the reference monitors**

Following sampling on the rooftop at Duke, we moved three PMS3003 units (labeled PMS3003-1 through -3 from the Duke University study site) to the US EPA AIRS on its RTP campus and further compared these three units to the more accurate



and precise regulatory FEMs (i.e., SHARP and two T640s). This allowed us to determine whether much of the poor performance of the Plantower PMS3003 sensors, the indistinct RH effects on the PMS3003 $PM_{2.5}$ measurements, and the unsuccessful temperature corrections to the PMS3003 $PM_{2.5}$ values, were attributable to the inferior precision of the E-BAM.

### 3.2.1 $PM_{2.5}$ concentration, RH, and temperature on 1 h scale

Table 1 indicates that the 1 h averaged ambient $PM_{2.5}$ levels at the US EPA RTP (9–10 μg m$^{-3}$) matched those at Duke University (9 μg m$^{-3}$). However, Fig. 5a shows 1 h time series data from all the reference monitors including the SHARP's embedded nephelometer and depicts smaller ranges of ambient $PM_{2.5}$ concentrations than were measured at Duke University. Table 1 indicates that the Std.Dev (less than 4 μg m$^{-3}$) and maximum $PM_{2.5}$ concentration (less than 20 μg m$^{-3}$) at the EPA RTP were significantly lower than at Duke University (9 μg m$^{-3}$ and 62 μg m$^{-3}$ for Std.Dev and maximum, respectively).
These comparisons imply that the RTP sampling location had overall lower ambient $PM_{2.5}$ concentrations and was consequently more challenging for low-cost sensors than the Duke University sampling site. During the measurement period, the mean RH and temperature were 64 ± 22% and 30 ± 7°C, respectively. The higher average RH level at the EPA RTP than at Duke University (45 ± 19%) accentuated the RH interference in the PMS3003 $PM_{2.5}$ measurements, as seen in Sect. 3.2.3.

### 3.2.2 PMS3003 performance characteristics on various timescales prior to adjustment for meteorological parameters

Figures 6a–b summarize graphically and statistically the pairwise correlations between all the instruments' 1 min aggregated and 1 h aggregated $PM_{2.5}$ mass concentrations, respectively. The $R^2$ and calibration factors between all the instruments on 1 min and 1 h scale were similar. The PMS3003 sensors were well correlated with one another ($R^2$ = 0.97), the two T640s ($R^2 \geq 0.63$) and the SHARP's embedded nephelometer ($R^2 \geq 0.49$) even for 1 min aggregated data at exceptionally low ambient $PM_{2.5}$ levels. In contrast, the 1 min or 1 h PMS3003–SHARP correlations ($R^2 \geq 0.25$) were poor and worse than the 1 h
PMS3003–E-BAM correlations ($R^2 \geq 0.36$) at the Duke site. Additionally, the SHARP had only moderate correlations with the two T640s ($R^2 \leq 0.58$) or the SHARP's embedded nephelometer ($R^2$ = 0.59) even though both the SHARP and T640 are US-designated $PM_{2.5}$ FEMs and the SHARP readings take into account its raw nephelometer values.

While the common optical-based principles of operation shared by T640 (and nephelometer) and PMS3003 could partially
explain the stark performance contrast between the SHARP and T640 (and nephelometer), the lower reported precision of the beta-attenuation-based approach with a 24 h average of ±2 μg m$^{-3}$ for SHARP than the T640 with an 1 h average of ±0.5 μg m$^{-3}$ in low ambient $PM_{2.5}$ concentration environments appears to be the root cause (Thermo Fisher Scientific, 2007; Teledyne Advanced Pollution Instrumentation, 2016). A previous study by Holstius et al. (2014) demonstrated the poor performance of BAM-1020 in a comparably low concentration environment in Oakland, CA. They have used both statistical
simulation based on the true ambient $PM_{2.5}$ distribution and the measurement uncertainty of BAM-1020 (1 h average: ±2.0–2.4 μg m$^{-3}$) provided by the manufacturer (Met One Instruments) and field test results to show that an $R^2$ of ~0.59 is as





correlated as one would expect from the 1 h measurements of a pair of collocated BAM-1020s. In contrast to the moderate intra-BAM-1020 correlation (~0.59) reported by Holstius et al. (2014), the two collocated T640s yielded an ideal $R^2$ of 0.95 (Fig. 6), which suggests a significantly smaller measurement error in the T640 than in the BAM-1020. The SHARP is known to derive its reported values by dynamically adjusting its embedded nephelometer readings based on its BAM measurements.

In other words, the SHARP performance was adversely affected by the low precision of its embedded BAM at low ambient $PM_{2.5}$ levels. All these observations seem to imply that beta-attenuation-based monitors might be unfavorable for low-cost particle sensor evaluation at the low concentrations typically present in the US. US EPA FEMs are valid for 24 h $PM_{2.5}$ measurements rather than for 1 h measurements (Jiao et al., 2016). An inappropriate selection of reference monitors might prejudice the overall performance of low-cost sensors particularly for time resolutions finer than 24 h.

The T640 sitting on the roof (T640_Roof) was chosen over the SHARP and the other T640 unit (T640_Shelter) as the reference monitor because 1) the T640 as a US-designated $PM_{2.5}$ FEM is better for sensor evaluation at low concentrations than a SHARP; 2) the T640_Roof had slightly lower correlations with the sensors than the T640_Shelter, therefore giving conservative estimates of PMS3003 performance. Figure 5b juxtaposes the T640_Roof and the three uncalibrated PMS3003

units $PM_{2.5}$ measurements at 1 h time resolution. Similar to the Duke University results, comparisons of the data using regression between the same set of instruments in Figs. 7a–d present similar calibration factors across the sensors on the same timescale, therefore indicating the excellent precision of the PMS3003 model. Unlike the analysis of the Duke University data, the calibration factors (prior to adjustments for meteorological parameters) varied little from one averaging timescale to another (Table 3). Despite an appreciable improvement in $R^2$ compared to the Duke University site being found

only on the 1 h scale, the accuracy of the T640 calibrated PMS3003 units substantially outperformed their E-BAM calibrated counterparts across the entire averaging time spectrum (Table 3) with the most pronounced difference on 1 h scale (27% vs. 201%). A less dramatic mean error drop from 1 h to 24 h scale at the EPA RTP (27% to 9%) compared to what was seen at the Duke University site (201% to 15%) highlights the inferior precision of the E-BAM and further undermines its credibility as a reference sensor at low $PM_{2.5}$ concentrations. It should be noted that the non-normally distributed residuals on 1 min, 1 h

and 6 h scales in Figs. 7a–c indicate that the true ambient $PM_{2.5}$ concentration term alone was not sufficient to explain the variation of PMS3003 measurements, therefore revealing the likely existence of RH or temperature impacts.

### 3.2.3 RH adjustment to sensor $PM_{2.5}$ measurements

As shown in Fig. 8, the empirical RH adjustment equation (i.e., Eq. (1)) fitted well with the 1 min, 1 h, and 6 h aggregated data ($R^2 \geq 0.48$). The regression fit statistics degraded when evaluating 12 h and 24 h aggregated data, likely because of an

insufficient number of observations and stronger smoothing effects at longer averaging time intervals. Aerosols at the EPA RTP generally exhibited smooth and continuous growth above the lowest collected RH rather than distinct deliquescence behavior (Fig. 8). The RH correction factors were roughly 20 to 30% above 1 even at the lowest RH (below 30%), which justifies the decision of conducting RH adjustments across the entire range of recorded RH without incorporating an RH



threshold. Despite the promising descriptions of correction factors as a function of RH, wide divergence in the magnitude of correction factors for a given RH exists. This divergence is likely the result of substantial day-to-day variation in the chemical composition of the aerosols (Day and Malm, 2000). A higher fraction of soluble inorganic compounds can contribute to a larger magnitude of RH correction factors (Day and Malm, 2000).

Figures 7e–g display the regressions of $PM_{2.5}$ measurements from the RH adjusted PMS3003 units versus the T640_Roof. The RH corrections brought the PMS–T640 correlations to above 0.90 for all 1 min, 1 h, and 6 h aggregated data. This significant improvement in $R^2$ implies a major RH influence that can explain up to nearly 30% of the variance in 1 min and 1 h PMS3003 $PM_{2.5}$ measurements in addition to the true ambient $PM_{2.5}$ concentration variable. Figure S3 demonstrates that the PMS3003-to-T640 ratios after the RH corrections were also considerably closer to a strict normal distribution than those with only the FEM corrections (Fig. S4). However, Figs. 7e–g suggest that the PMS3003 $PM_{2.5}$ measurements were still not in complete agreement with the T640 readings even after the RH adjustments. This discrepancy might stem from variations in aerosol composition described previously or impacts of particle size biases (Chakrabarti et al., 2004), therefore warranting a further step of FEM conversion (adjustment). According to Table 3, the combination of RH and FEM corrections were able to substantially improve the accuracy of PMS3003 $PM_{2.5}$ measurements by reducing the mean errors to within 12% even for data at 1 min time resolution. The ideal normal distribution of PMS3003-to-T640 ratios in combination with the high accuracy and precision of the finest-grained data proves especially beneficial for minimization of exposure measurement errors in short-term $PM_{2.5}$ health effect studies (Breen et al., 2015) or mapping of intra-urban $PM_{2.5}$ exposure gradients (Zimmerman et al., 2018).

20 **3.2.4 Temperature adjustment to sensor $PM_{2.5}$ measurements**

The decision to conduct the temperature adjustments to 1 min, 1 h, 6 h, and 12 h aggregated PMS3003 $PM_{2.5}$ measurements was based on the AIC results in Table S1. Table S1 demonstrates that the AIC values of the calibration models incorporating an additional temperature term were substantially lower than those of the models including only a true $PM_{2.5}$ concentration term at these levels of temporal resolution, therefore indicating the significance of the temperature variable in the calibration models. The 24 h AIC values are not reported as 24 h observations generally have limited statistical power to determine the significance of temperature in the models.

As shown in Table 3, the temperature corrections (when available) could further reduce the mean PMS3003 $PM_{2.5}$ measurement errors by no more than 4%, with the largest reduction in mean errors found in the 12 h averaged data. This marginal improvement achieved stands in marked contrast to that brought about by the RH corrections (up to 17%), suggesting the triviality of temperature adjustments in the entire suite of calibrations. Nevertheless, the addition of the temperature adjustments succeeded in lowering the mean errors to within 10% at 1 min, 1 h, and 6 h time resolutions, which were comparable to the value at 24 h time resolution (9%). Figure S5d also depicts the PMS3003-to-T640 ratios at 12 h





averaging interval after the temperature corrections and shows that these ratios were slightly more normally distributed than those with only the FEM corrections (Fig. S4). As a result, whether to conduct temperature adjustments is contingent upon the error targets, which are further dependent on the performance goals for the desired applications.

### 3.3 IIT Kanpur high ambient PM$_{2.5}$ concentration environment with E-BAM as the reference monitor

Low-cost particle sensors are commonly known to exhibit an upward trend in accuracy with increasing ambient PM$_{2.5}$ concentrations (Williams et al., 2017; Johnson et al., 2018). Moreover, Kanpur presents distinct seasonal variations in the particle size distribution. During the early stage of the monsoon season (June), coarse mode aerosols are predominant due to the transport of dry dust particles from the western Thar Desert or arid regions to Kanpur. In contrast, during the post-monsoon season, anthropogenic accumulation mode aerosols transported from the north and northwest dominate over Kanpur (Sivaprasad and Babu, 2014; Li et al., 2015; Bran and Srivastava, 2017). We explored how the variability in the ambient PM$_{2.5}$ concentrations and the particle size distribution affected the low-cost PM sensors' performance and calibration curves relative to the reference monitor (E-BAM in our study).

### 3.3.1 PM$_{2.5}$ concentration, RH, and temperature on 1 h scale

Table 1 shows that Kanpur had significantly higher ambient PM$_{2.5}$ levels for a 1 h averaging period during the post-monsoon season (116 ± 57μg m$^{-3}$) than during the monsoon season (36 ± 17μg m$^{-3}$). This seasonal increase in ambient PM$_{2.5}$ concentrations is aligned with our expectation and can be attributed to diminished wet scavenging by precipitation, a shallow boundary layer (mixing height), and lower ventilation coefficients (wind speed) during the post-monsoon season (Gaur et al., 2014). While only moderately high ambient PM$_{2.5}$ levels were found during the Kanpur monsoon season, they were substantially higher than those measured at the Duke University site (9 ± 9μg m$^{-3}$). The field tests in this study provided a wide range of ambient PM$_{2.5}$ levels spanning from high (Kanpur post-monsoon season), moderate (Kanpur monsoon season), to low (Duke University site). This PM$_{2.5}$ concentration range coupled with the same type of reference monitor (E-BAM) is ideal for constructing empirical error curves to investigate the sensor performance within each individual concentration class as a function of averaging time period (as discussed in Sect. 3.3.4). The RH values during the monsoon season (62 ± 15%) were comparable to those during the post-monsoon season (63 ± 16%). These RH values collected in Kanpur were also similar to those at the EPA RTP site (64 ± 22%). The temperature during the monsoon season (33 ± 5°C) was considerably higher than that during the post-monsoon season (22 ± 4°C).

### 3.3.2 Comparing calibrations across locations

As with the two field tests in the low concentration region, the two PMS3003 units were highly correlated with each other during both the monsoon (R$^2$ = 0.99) and post-monsoon seasons (R$^2$ = 0.93) in Kanpur (Fig. S6). This good agreement is also reflected in Fig. 9, which displays that the two sensors were in sync and tracked reasonably well with the E-BAM. However, there was a minor decrease in the intra-sensor correlation from the monsoon to post-monsoon seasons that might signal a



performance change of the two PMS3003 sensors either due to minor deterioration or a change in the pollutant source. Figure S6 illustrates that the magnitude of the deviation from the regression line during the monsoon season was likely irrelevant to the deployment time (measured by the number of hours past the beginning of the Kanpur study, i.e., 2017 June 08 00:00). In contrast, the extent of the divergence was somewhat larger for the longer deployment time near the high end of

the $PM_{2.5}$ range over the post-monsoon period. One plausible explanation for the distinguishable post-monsoon (but not monsoon season) change is the routine exposure (for nearly a month) of the sensors to high concentrations of accumulation mode aerosols. This may be especially detrimental to PM sensors; all the more so because the foggy condition during post-monsoon and winter over Kanpur may further exacerbate the accumulation of aerosol particles at lower surfaces and therefore the deposition of particles within the sensors (Li et al., 2015; Bran and Srivastava, 2017). This constant exposure

possibly caused disproportionately large detection errors primarily near the upper end of the $PM_{2.5}$ range. Another possible explanation is the change of dominant pollutant source from the early stage of monsoon (long-range transport of mineral dust from Iran, Afghanistan, Pakistan, and the Thar Desert) to post-monsoon (local impact of biomass burning emissions) season (Ram et al., 2010). Sensors are likely to respond differently to different varieties of aerosols and the change in sensor responses might be most pronounced near the upper end of the $PM_{2.5}$ range. Figure 9b substantiates the potential change by

showing that the two uncalibrated PMS3003s were unable to match the troughs of the E-BAM (even troughs below 40 µg m$^{-3}$) throughout the post-monsoon season, as they were during the monsoon season in Fig. 9a.

Despite the slight potential change, higher PMS3003–E-BAM correlations were found in the post-monsoon season than the monsoon season over all time averaging intervals (Table 4). Figure 10 displays the 1 h and 24 h average regression plots for

the two uncalibrated sensors against the E-BAM during the monsoon and post-monsoon seasons. Similar to the Durham and EPA RTP field tests, different PMS3003 units had similar calibration factors over the same averaging timescales during both seasons. Comparable to the EPA RTP evaluation, the sensor units at or in the same study location or season were roughly similar in sensitivity and baseline regardless of averaging time periods (Fig. 10 and Table 4). Figure 10 also shows a distinct baseline drift of the PMS3003s from the monsoon to the post-monsoon season regime. This appreciable drift in baseline

agreed with the sensors being incapable of reaching the troughs of true ambient $PM_{2.5}$ concentrations. This may also suggest a performance change or may be a reflection of a different calibration regime.

Figure 11 depicts a heat map of mean errors in calibrated PMS3003 $PM_{2.5}$ measurements with respect to averaging timescales and calibration methods across varied sampling locations or seasons. Even though the EPA RTP sampling

location had the lowest ambient $PM_{2.5}$ level among the three study locations, it achieved the highest accuracy over each averaging time period, therefore reiterating a vital role the precision of reference instruments plays in evaluating sensor performance. For the remaining two sampling sites with an E-BAM as the reference monitor, lower errors were generally found in higher $PM_{2.5}$ concentration environments. The exceptions to this rule were observed at 12 h (Kanpur post-monsoon error > monsoon error) and 24 h (Kanpur monsoon error > Duke University site error) time intervals. The occurrence of



these anomalies can be explained by stronger smoothing effects than $PM_{2.5}$ concentration effects over longer averaging times. Table 4 details the errors in calibrated PMS3003 $PM_{2.5}$ measurements during the monsoon and post-monsoon seasons in Kanpur. The appreciably narrower reductions in mean errors from 1 h to 24 h scale during both seasons in Kanpur (monsoon: 46% to 17%, post-monsoon: 35% to 11%) compared to the reduction at Duke University site (201% to 15%)

underscore the inferior precision of E-BAM at low ambient $PM_{2.5}$ concentrations.

The lack of requirement for RH corrections during both testing seasons in Kanpur paralleled the outcomes of the Duke University field test. Figure S7 shows that the empirical RH correction equation fitted poorly with the widely scattered data from both monsoon ($R^2 \le 0.13$) and post-monsoon seasons ($R^2 \le 0.03$). We speculate that the E-BAM's low precision might

be responsible for the failure to establish the impact of RH on PMS3003 responses, considering that the T640 measurements resulted in a significant RH relationship under similar conditions. We attempted to apply the empirical RH adjustment equations derived at the EPA RTP testing site to the Kanpur and Duke University data sets. However, no improvements in correlations or errors were found, indicating RH correction function appears to be highly specific to study sites because of its great reliance on particles' chemical, microphysical, and optical properties (Laulainen, 1993). The temperature variable was

found statistically significant and therefore incorporated in the calibration models at time resolutions finer than 6 h for the Kanpur monsoon data, and finer than 12 h for the post-monsoon data (Table S1). Overall, the temperature adjustments can scale the PMS3003 $PM_{2.5}$ measurement errors down by no more than 7%, with the 6 h averaged data during the post-monsoon season marking the greatest improvement (Table 4). These marginal improvements were comparable to those observed at the EPA RTP testing site (within 4%).

**3.3.3 Comparing between the methods for calibrating the Kanpur post-monsoon measurements**

We observed a relatively pronounced non-linear relationship between the raw PMS3003 and the E-BAM $PM_{2.5}$ responses over the full concentration range examined during the post-monsoon season at IIT Kanpur (Fig. 10). In previous research, similar nonlinearity was ubiquitously characterized by attenuated responses towards the upper end of low-cost sensors' operation range in both field campaigns (Gao et al., 2015; Kelly et al., 2017; Johnson et al., 2018) and laboratory settings

(Austin et al., 2015; Wang et al., 2015). The shape of calibration curves is dependent on varied factors such as type of low-cost sensor, range of true ambient $PM_{2.5}$ concentrations, particle size and particle composition (Wang et al., 2015). Without additional information, we are unable to parse out the exact reasons for the occurrence of this nonlinearity in our data during the Indian post-monsoons season. Nevertheless, we speculate that the comparatively high concentration range along with the prevalence of small particles encountered during the post-monsoon season might account for this nonlinearity (Kelly et al.,

2017). In the present study, the PMS3003 responses were well characterized by a linear model below ~125 μg m$^{-3}$, which was close to the highest 1 h $PM_{2.5}$ concentration during the monsoon season. This threshold was around 3 times greater than that reported by Kelly et al. (2017), who field-tested PMS1003s under an ammonium nitrate dominated, moderately high $PM_{2.5}$ concentration condition (1 h $PM_{2.5}$ mean: up to 20 μg m$^{-3}$, range: 10–70.6 μg m$^{-3}$).




Researchers have used higher-order polynomial (Austin et al., 2015; Gao et al., 2015), penalized spline (Austin et al., 2015), and exponential functions (Kelly et al., 2017) to capture non-linear responses of low-cost sensors. In this study, we explored the quadratic model to describe the full range response of the PMS3003s during the Kanpur post-monsoon season. The

quadratic model was chosen because it is straightforward to understand, interpret, disseminate, and use. The time series of the 1 h and 24 h averages of the calibrated PMS3003 PM$_{2.5}$ responses using the two calibration models (i.e., simple linear and quadratic models) can be found in Fig. S8. Figure S8 shows that the quadratic model might suit the post-monsoon 1 h aggregated data better than the simple linear model as the simple linear model failed to capture the troughs of the E-BAM throughout the post-monsoon period. The two models only differed little for the 24 h aggregated data. This is expected as

Fig. 10 and Fig. S9 display that the strength of nonlinearity declined as the averaging times increased because longer averaging times reduced the number of relatively low concentration observations (such as below ~100 µg m$^{-3}$). Table S2 summarizes the goodness of fit and accuracy estimates for the two model types as a function of time averaging intervals during the post-monsoon season. Table S2 indicates that the quadratic fit appeared to have better goodness of fit and accuracy estimates for the current post-monsoon data set than the simple linear fit with both lower AIC and RMSE values at

all time resolutions. Compared to the simple linear model, the quadratic model could further improve the mean accuracy of PMS3003 PM$_{2.5}$ responses by up to 11% (Table 4). Even when the nonlinearity was not strong enough to make the simple linear fit statistically different from the quadratic fit (i.e., the quadratic term $a_2$ in the quadratic fit (Eq. (7)) not significantly different from 0 with p>0.1) at 24 h integration time, the quadratic fit can still reduce the mean error and RMSE by 2% (Table 4) and 2 µg m$^{-3}$ (Table S2), respectively. This might also shed some light on the choice of calibration methods for

PMS3003 PM$_{2.5}$ responses in future deployments. The quadratic model should be chosen over the simple linear model as the starting point (default approach) to PMS3003 PM$_{2.5}$ responses calibration since the quadratic model can always be of larger benefit to the accuracy of PMS3003 measurements than the simple linear model even when the nonlinearity is weak at low ambient PM$_{2.5}$ concentrations or at longer time averaging intervals.

### 3.3.4 Empirical error curves for PMS3003 PM$_{2.5}$ measurements with E-BAM as the reference monitor

Empirical error curves for PMS3003 PM$_{2.5}$ measurements by calibration method and averaging time are presented in Fig. 12 by combining the results of all the field tests with E-BAM as the reference monitor (i.e., Duke University and IIT Kanpur data sets). These curves are useful for easy reference to the magnitude of errors for a given concentration range at a given temporal resolution. Overall, regardless of the averaging times, the largest errors were found below 20 µg m$^{-3}$, particularly in the range of 0 to 10 µg m$^{-3}$. Although further work is required to improve the error curves by collecting more data points

especially near the upper end of the PM$_{2.5}$ distributions, we would presume calibrated PMS3003 PM$_{2.5}$ responses to be relatively stable and consistent above ~70 µg m$^{-3}$ for 1 h aggregated data and above ~50 µg m$^{-3}$ for 6 h to 24 h aggregated data with uncertainties roughly confined within 25%, particularly when the quadratic calibration models are employed.




Given the broad range in PM$_{2.5}$ concentrations, Fig. 12 seems to demonstrate that the quadratic calibration method performed better than their simple linear counterpart at all time intervals with steadier mean of ratios lines (remaining more constantly at 1 regardless of concentration classes) and relatively low uncertainties. The quadratic model outperformed the simple linear model particularly over the moderately high concentration range (i.e., ~60–140 µg m$^{-3}$). Although a lesser improvement than over the moderately high concentration range, the quadratic fit still managed to slightly tighten the shaded uncertainty region over the range of ~30–60 µg m$^{-3}$, where few differences existed between the two calibration curves. Table S3 shows that the quadratic fit had smaller AIC and RMSE values than the simple linear fit at all time intervals. Figure S10 further shows that the quadratic models fitted remarkably better than the simple linear model to the data. These observations support using the quadratic rather than the simple linear method as the general approach in calibrating PMS3003 PM$_{2.5}$ responses.

## 4 Conclusions

This study comprised three distinct field campaigns in both an urban-influenced setting in Kanpur, India during both monsoon (1 h averages: [PM$_{2.5}$] = 36 ± 17µg m$^{-3}$; RH = 62 ± 15%; temperature = 33 ± 5°C) and post-monsoon seasons ([PM$_{2.5}$] = 116 ± 57µg m$^{-3}$; RH = 63 ± 16%; temperature = 22 ± 4°C) and two suburban settings in Durham ([PM$_{2.5}$] = 9 ± 9µg m$^{-3}$; RH = 45 ± 19%; temperature = 15 ± 8°C) and RTP, NC, US ([PM$_{2.5}$] = 10 ± 3µg m$^{-3}$; RH = 64 ± 22%; temperature = 30 ± 7°C). The goal is to provide the adequate range of conditions to characterize how variability in ambient PM$_{2.5}$ concentrations, meteorological factors (such as temperature and RH), and reference monitor types (Durham and Kanpur: E-BAM; RTP: T640 and SHARP) can affect the performance of low-cost Plantower PMS3003 sensors' PM$_{2.5}$ measurements against reference instruments at 1 min, 1 h, 6 h, 12 h and 24 h integration times. This information is ultimately important for identifying suitable research or citizen science applications for these sensors given their quantified capabilities.

The lower mean errors of PMS3003s at the EPA RTP site (from 27% for 1 h to 9% for 24h) than those at the remaining sites (Duke: from 201% to 15%; Kanpur monsoon: from 46% to 17%; Kanpur post-monsoon: from 35% to 11%) underscores the critical role the precision of reference instruments (T640: ±0.5 µg m$^{-3}$ for 1 h; SHARP: ±2 µg m$^{-3}$ for 24 h, better than the E-BAM) plays in evaluating sensor performance and the potential unfavorability of beta-attenuation-based monitors for testing sensors at low concentrations. Nonetheless, longer averaging times (such as 24 hours) typically smoothed out noisy signals and resulted in similar levels of error, indicating the feasibility of calibrating sensors using suboptimal reference analyzers as long as an appropriate averaging time is chosen. Even though the RH correction factor models might be highly location-specific, it is striking to see that they were capable of explaining up to nearly 30% of the variance in 1 min, 1 h and 6 h aggregated sensor measurements and reducing mean errors down from ~22–27% to roughly 10% even at the finest 1 min time resolution. Compared to the RH corrections, temperature corrections were found to be relatively small and can only scale uncertainties down by 7% at most; however, in addition to the other corrections this may help to achieve the highest possible accuracy level. It is important to note that the success of both RH and temperature corrections relies on the precision





of reference instruments. Additionally, we observed that PMS3003s exhibited non-linear PM$_{2.5}$ responses relative to an E-BAM when ambient PM$_{2.5}$ levels exceeded ~125 µg m$^{-3}$. We found that the quadratic model is more suitable than the simple linear regression model for effectively capturing this nonlinearity and can further reduce mean errors by up to 11%. Furthermore, we demonstrated that the quadratic model should be chosen over the simple linear model as the starting point (default approach) in calibrating PMS3003 PM$_{2.5}$ responses since the quadratic model can always be of larger benefit to the accuracy of PMS3003 measurements than the simple linear model even when the nonlinearity is weak at low ambient PM$_{2.5}$ concentrations or at longer time averaging intervals. The empirical error curves constructed by pooling the results of all the field tests with E-BAMs as the reference monitor were indicative of relatively stable and consistent calibrated responses above ~70 µg m$^{-3}$ for 1 h aggregated data and above ~50 µg m$^{-3}$ for 6 h to 24 h aggregated data with uncertainties roughly confined within 25%, particularly when the quadratic calibration models are employed.

Overall, we conclude that the Plantower PMS3003 sensors, as a promising low-cost PM monitor, can achieve high accuracy and precision over a wide range in PM$_{2.5}$ concentration, but only after applying appropriate calibration models using ideal reference monitors and after adjusting for meteorological parameters. The insights gleaned from this study suggest that establishing dense, wireless, real-time PM sensor networks in hazy urban areas such as Delhi and Mumbai, India to approximate the location of major PM$_{2.5}$ sources (local vs. regional) and to better understand the influence of meteorology such as specific wind patterns on the resulting regional PM$_{2.5}$ levels in order to guide local and regional air quality management (Hagler et al., 2006) is feasible with current low-cost sensing technology with proper calibrations.

**Data availability**

The data are available upon request to Tongshu Zheng (tongshu.zheng@duke.edu).

**Competing interests**

The authors declare that they have no conflict of interest.

**Disclaimer**

The US Environmental Protection Agency (EPA) through its Office of Research and Development participated in this research. The views expressed in this paper are those of the authors and do not necessarily reflect the views or policies of EPA. It has been subjected to EPA Agency review and approved for publication. Mention of trade names or commercial products does not constitute endorsement or recommendation for use.





**Acknowledgments**

Sachchida N. Tripathi, Shilpa Shirodkar, and Ronak Sutaria are supported under the Research Initiative for Real-time River Water and Air Quality Monitoring program funded by the Department of Science and Technology, Government of India and Intel Corporation, and administered by the Indo-US Science and Technology Forum. The authors would like to thank Tim
Hanley at EPA OAQPS for providing the raw 1 min data from the T640 PM mass monitors used in the current study. The authors are also grateful to Christina Norris at Duke University for her insightful advice and comments on the paper.

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



Figure 1: (a) The custom-designed printed circuit board (PCB) and its components for the Plantower PMS3003 sensor packages. (b) Electrical box housing all components for outdoor sampling.




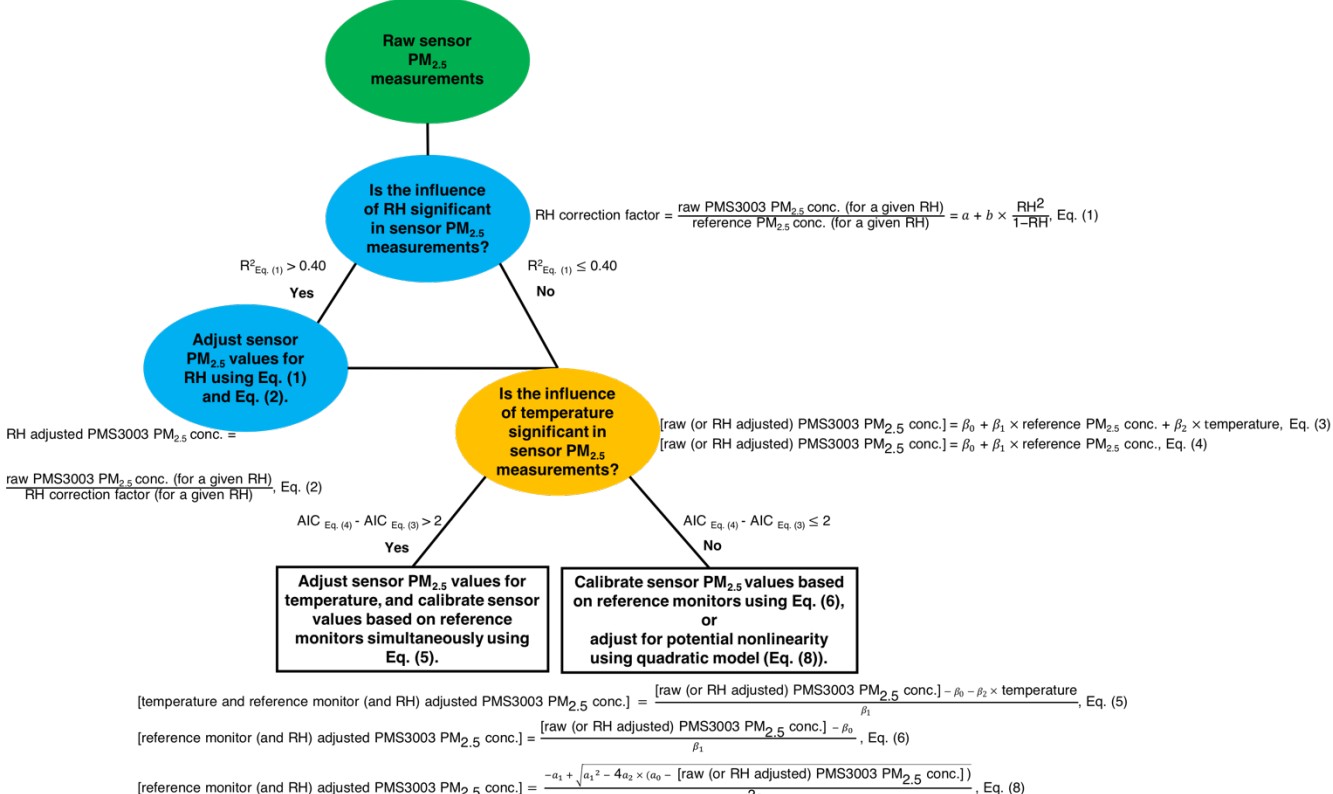

**Figure 2: Flow path for sensor calibrations.**





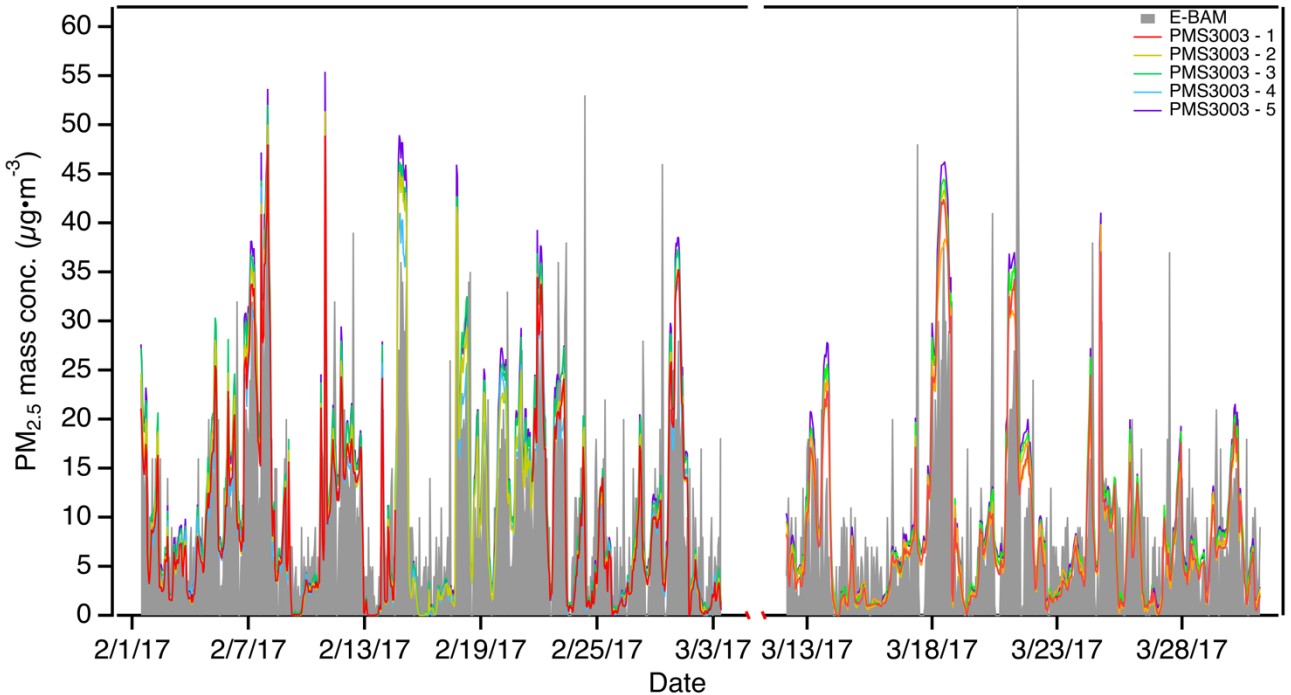

**Figure 3: Comparison of hourly PM$_{2.5}$ mass concentrations between the E-BAM and the five uncalibrated PMS3003 sensor packages between February 1, 2017 and March 31, 2017 at Duke University.**

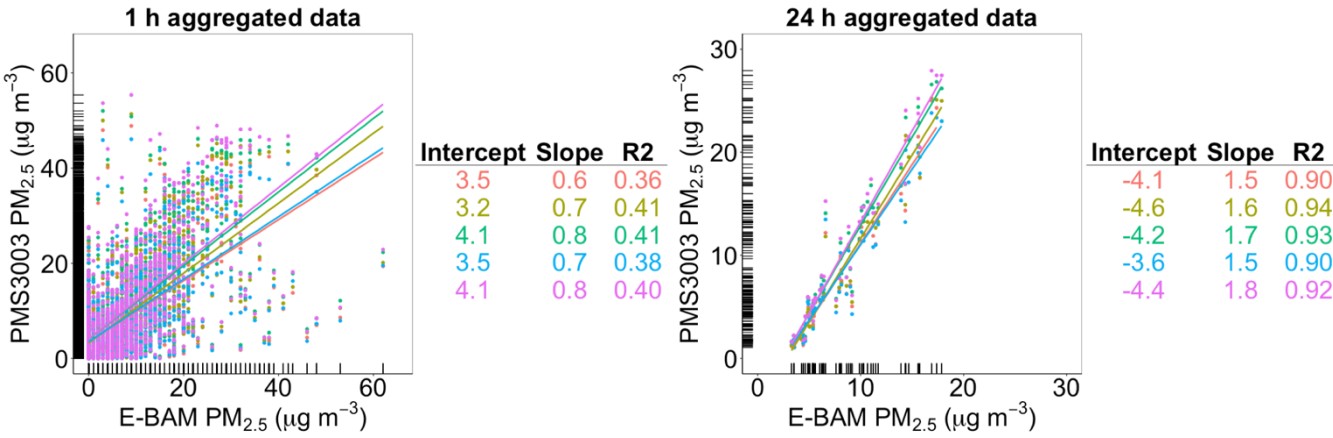

**Figure 4: Linear regressions between aggregated PM2.5 mass concentrations (µg m$^{-3}$) of the E-BAM and the five uncalibrated PMS3003s at 1 h and 24 h time intervals from February 1, 2017 to March 31, 2017 at Duke University (6 h and 12 h results not shown). Marginal rugs were added to better visualize the distribution of data on each axis.**



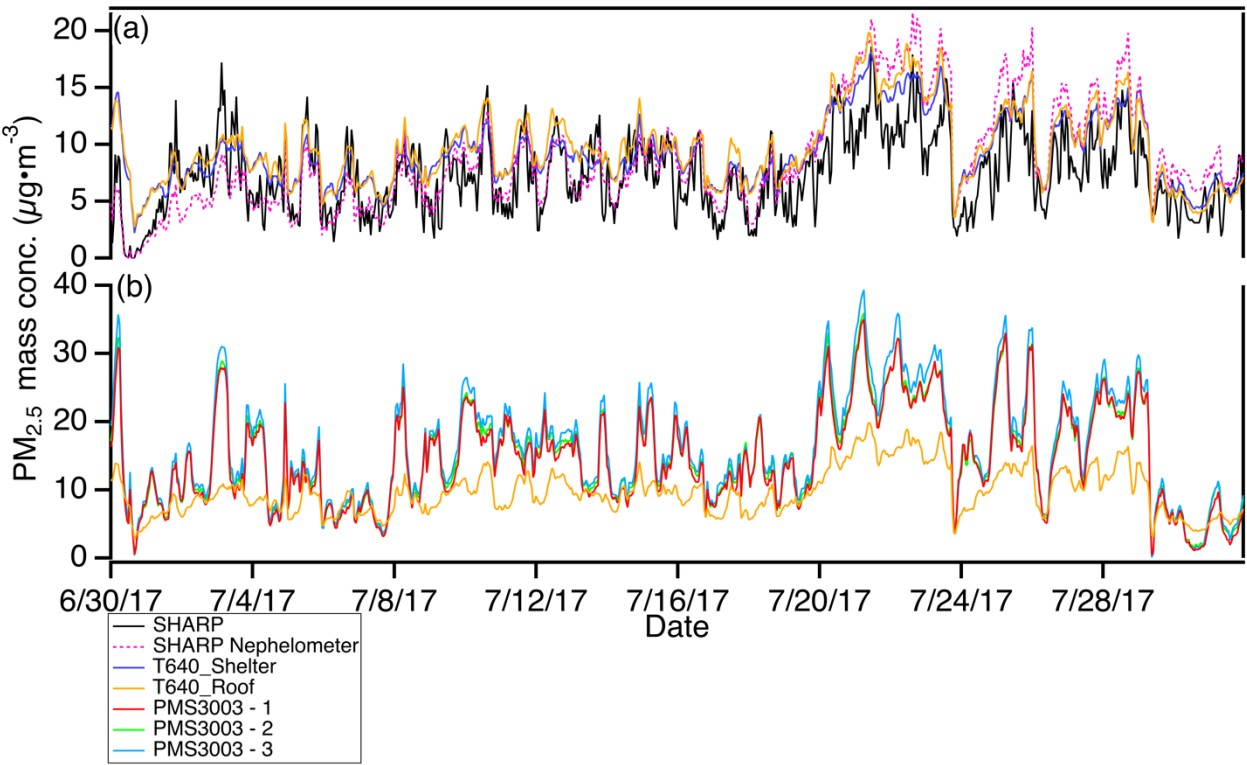

**Figure 5: Comparison of hourly aggregated PM$_{2.5}$ mass concentrations (in µg m$^{-3}$) a) between the SHARP, the SHARP's nephelometer, the two T640s (one unit sitting on the roof "T640_Roof", the other unit installed in the OAQPS shelter "T640_Shelter"), from June 30, 2017 to July 31, 2017 at US EPA RTP, b) between the T640 sitting on the roof (T640_Roof) and the three uncalibrated PMS3003 sensor packages during the same period at the same location.**





**Figure 6:** Pairwise correlations between (a) 1 min aggregated PM$_{2.5}$ mass concentrations (μg m$^{-3}$) (b) 1 h aggregated PM$_{2.5}$ mass concentrations (μg m$^{-3}$) of the SHARP, the SHARP's nephelometer, the two T640s, and the three uncalibrated PMS3003 sensor packages between June 30, 2017 and July 31, 2017 at US EPA RTP. In both (a) and (b), the upper-right set of panels includes the intercept, slope, and R$^2$ of linear regression models using the ordinary least squares (OLS) method; the lower-left set of panels shows the linear regression lines superimposed on pairwise plots.





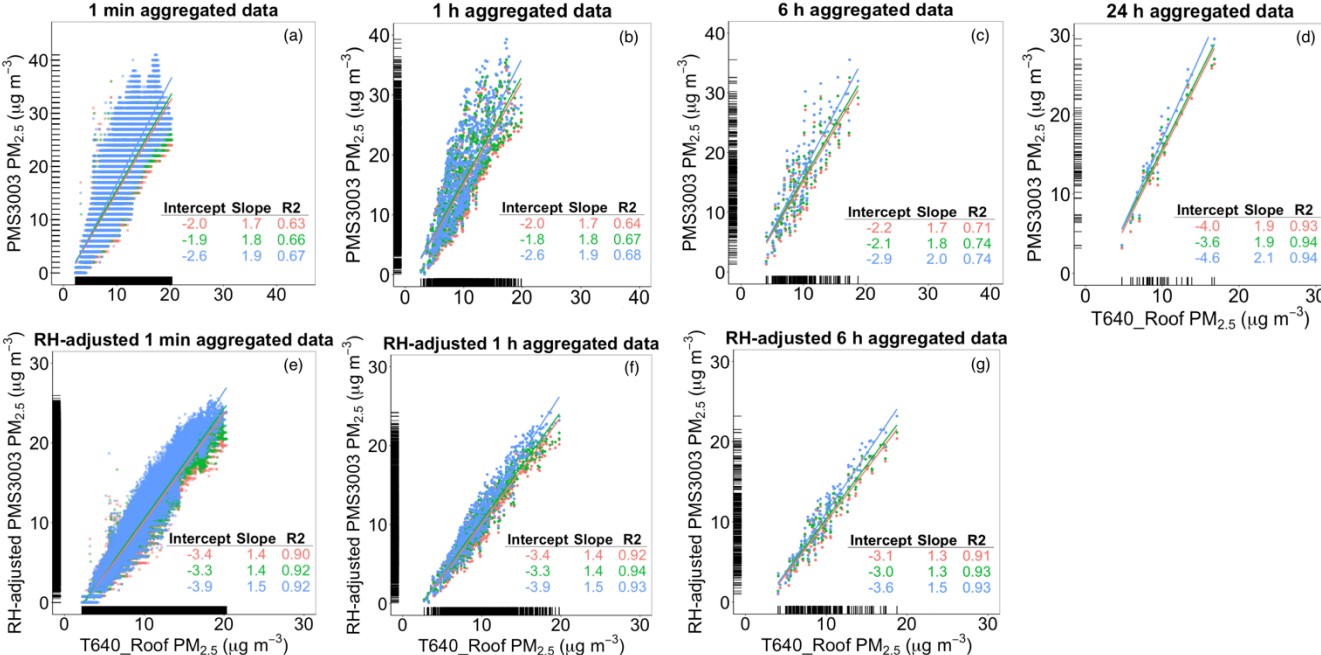

**Figure 7: Linear regressions between aggregated PM$_{2.5}$ mass concentrations (µg m$^{-3}$) of the T640 sitting on the roof (T640_Roof) and the three PMS3003s from June 30, 2017 to July 31, 2017 at US EPA RTP. In a–d, the PMS3003 readings are raw values at 1 min, 1 h, 6 h, and 24 h, respectively (12 h results are not shown). In e–g, the PMS3003 readings are RH-adjusted values at 1 min, 1 h, and 6 h, respectively. Marginal rugs were added to better visualize the distribution of data on each axis. Note the rug on the y axis in a is sparse because 1 min raw PMS3003 PM$_{2.5}$ measurements are recorded as integers.**

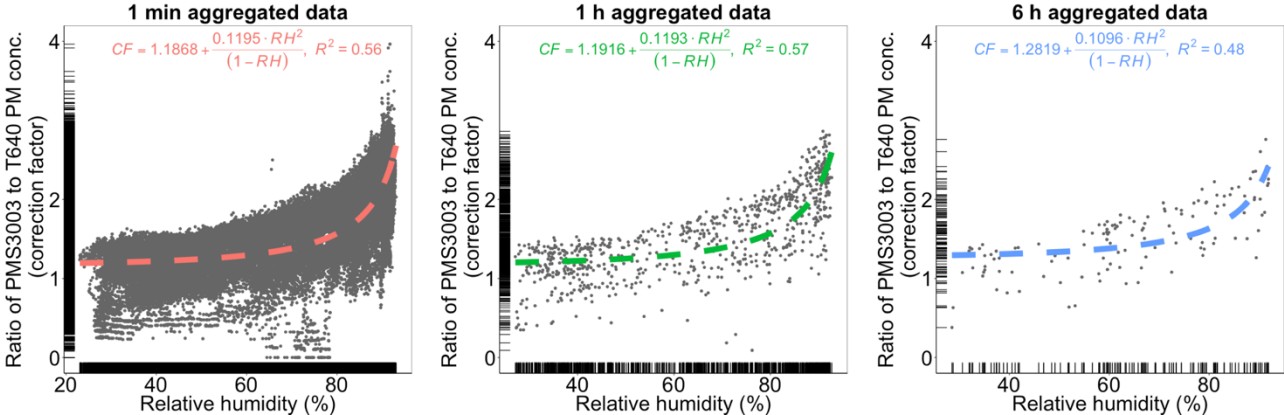

**Figure 8: Fractional increase in PM$_{2.5}$ weight measured by the uncalibrated PMS3003 sensors with respect to RH at 1 min, 1 h, and 6 h time intervals from June 30, 2017 to July 31, 2017 at US EPA RTP. RH (%) and PMS3003 PM$_{2.5}$ concentrations (µg m$^{-3}$) are arithmetic means averaged across all the three PMS3003 sensor packages at each point in time. The fitted RH adjustment equations and curves were superimposed on the plots. Marginal rugs were added to better visualize the distribution of data on each axis. The results of 12 h and 24 h aggregated data are not shown as their patterns are relatively indistinct.**







**Figure 9: Comparison of hourly PM₂.₅ mass concentrations between the E-BAM and the two uncalibrated PMS3003 sensor packages a) from June 8, 2017 to June 29, 2017 (monsoon season), and b) from Oct 23, 2017 to Nov 16, 2017 (post-monsoon season) at IIT Kanpur.**





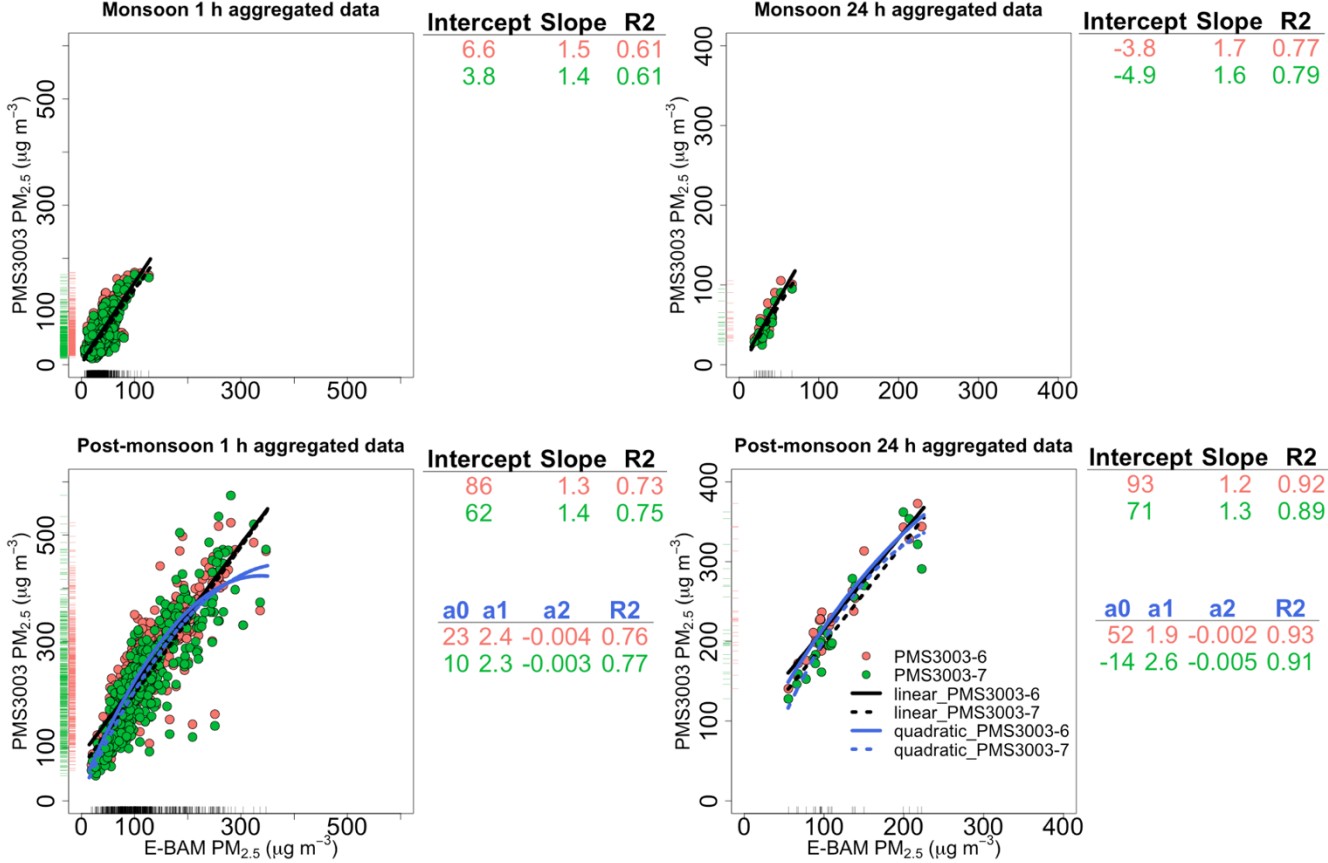

**Figure 10: Linear regressions between aggregated PM$_{2.5}$ mass concentrations (µg m$^{-3}$) of the E-BAM and the two uncalibrated PMS3003s at 1 h and 24 h time intervals during the monsoon season (from June 8, 2017 to June 29, 2017), and the post-monsoon season (from Oct 23, 2017 to Nov 16, 2017) at IIT Kanpur (6 h and 12 h results are shown in Fig. S9). The fit coefficients for the calibration models are provided. Marginal rugs were added to better visualize the distribution of data on each axis.**





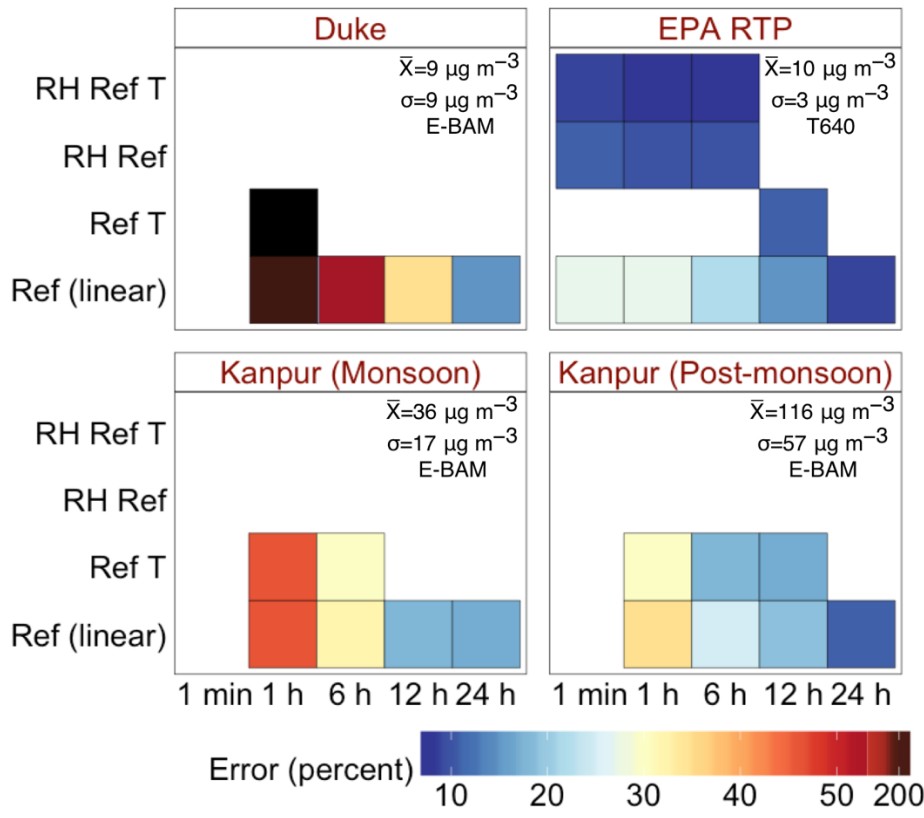

**Figure 11: Heat map of the mean errors of the calibrated PMS3003 PM$_{2.5}$ measurements with respect to averaging timescales and calibration methods across study sites or sampling seasons. The mean and Std.Dev of the true ambient PM$_{2.5}$ concentrations reported by the corresponding reference instrument (Ref) for each location or season were overlaid on the heat map. Note the errors of the 1 h E-BAM calibrated, and the combination of E-BAM and temperature (T) calibrated PMS3003 PM$_{2.5}$ measurements at the Duke study site were 201% and 207%, respectively. They are represented by dark brown and black, respectively to improve the visual contrast in errors.**





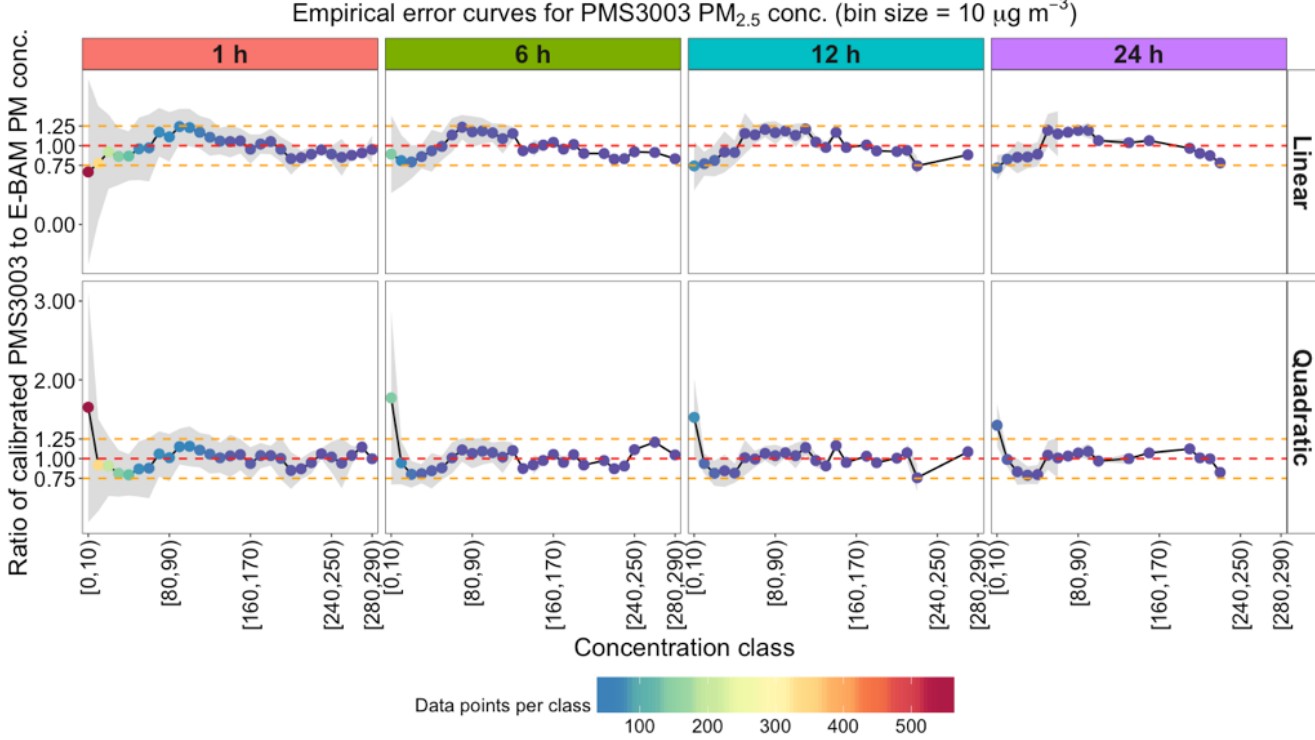

**Figure 12: Empirical error curves for the E-BAM calibrated Plantower PMS3003 PM$_{2.5}$ measurements at 1 h, 6 h, 12 h, and 24 h time intervals by two different calibration methods (i.e., simple linear and quadratic equations). The curves were generated from the combination of the Duke University and IIT Kanpur data sets. The points and lines represent the means of ratios of E-BAM-calibrated-PMS3003 to E-BAM PM$_{2.5}$ measurements in different concentration classes, each of which spans a 10 μg m$^{-3}$ interval. The shaded region represents the corresponding magnitudes of errors of PMS3003 PM$_{2.5}$ measurements after the E-BAM calibration. The concentration classes are color coded by the number of data points in each class. Note the shaded region is generally absent from near the upper end of the PM$_{2.5}$ ranges due to insufficient observations for the error evaluation. The red dashed line indicates ratio of 1, while the two orange dashed lines indicate ratio of 0.75 and 1.25, respectively.**



**Table 1: Summary statistics for 1 h averaged measurements [mean ± Std.Dev (range)] at the three sampling locations. Reference monitors at the sampling locations are indicated with shading.**

| Location | Date | Instruments | PM$_{2.5}$ (µg m$^{-3}$) | RH (%) | Temperature (°C) | Data completeness |
|---|---|---|---|---|---|---|
| Duke rooftop (36.003350˚ N, 78.940259˚ W) | 2/1/2017– 3/31/2017[*] | PMS3003-1 | 9±9 (0–49) | | | 86% |
| | | PMS3003-2 | 10±10 (0–51) | | | 100% |
| | | PMS3003-3 | 11±10 (0–52) | | | 100% |
| | | PMS3003-4 | 9±9 (0–46) | | | 100% |
| | | PMS3003-5 | 11±11 (0–55) | | | 100% |
| | | **E-BAM** | **9±9 (0–62)** | | | **100%** |
| | | Average Sparkfun SHT15 | | 45±19 (9–87) | 15±8 (0–36) | 100% |
| US EPA RTP (35.882816˚ N, 78.874471˚ W) | 6/30/2017– 7/31/17 | PMS3003-1 | 15±7 (0–35) | | | 100% |
| | | PMS3003-2 | 15±7 (0–36) | | | 100% |
| | | PMS3003-3 | 16±8 (0–39) | | | 100% |
| | | **SHARP** | **7±4 (0–19)** | | | **99%** |
| | | SHARP Nephelometer | 9±5 (0–22) | | | 99% |
| | | **T640_Roof** | **10±3 (3–20)** | | | **100%** |
| | | **T640_Shelter** | **9±3 (2–18)** | | | **100%** |
| | | Average Sparkfun SHT15 | | 64±22 (27–93) | 30±7 (14–45) | 100% |
| IIT Kanpur rooftop (26.515818˚ N, 80.234337˚ E) | 6/8/2017– 6/29/17 (monsoon) | PMS3003-6 | 55±31 (7–173) | | | 100% |
| | | PMS3003-7 | 49±29 (7–170) | | | 100% |
| | | **E-BAM** | **36±17 (0–127)** | | | **85%** |
| | | Weather station | | 62±15 (30–88) | 33±5 (24–43) | 93% |
| | 10/23/2017– 11/16/17 (post-monsoon) | PMS3003-6 | 237±88 (57–523) | | | 98% |
| | | PMS3003-7 | 219±91 (47–574) | | | 98% |
| | | **E-BAM** | **116±57 (19–347)** | | | **93%** |
| | | Weather station | | 63±16 (19–88) | 22±4 (14–35) | 99% |

[*]All the PMS3003 sensor packages and the E-BAM were shut down between March 3 and March 12 for maintenance.



**Table 2: Summary of sensor performance characteristics for the five PMS3003 PM$_{2.5}$ measurements at 1 h, 6 h, 12 h, and 24 h time intervals from February 1, 2017 to March 31, 2017 at Duke University. The fit coefficients for the calibration models are provided. The R$^2$, mean of ratios, and error are performance characteristics for the calibrated sensor PM$_{2.5}$ measurements in comparison with reference values. The results are displayed in mean (range) format. Note the mean statistics were obtained by fitting the models to the PMS3003 PM$_{2.5}$ measurements averaged across all five sensor package units at each point in time.**

| Performance characteristics | 1 h | | 6 h | 12 h | 24 h |
|---|---|---|---|---|---|
| adjustment | E-BAM | E-BAM, T | E-BAM | E-BAM | E-BAM |
| $\beta_0$ | 3.7 (3.2–4.1) | 4.5 (4.1–5.1) | -1.9 (-2.3--1.4) | -2.4 (-2.8--1.8) | -4.2 (-4.6--3.6) |
| $\beta_1$ | 0.7 (0.6–0.8) | 0.7 (0.7–0.8) | 1.4 (1.2–1.5) | 1.4 (1.3–1.5) | 1.6 (1.5–1.8) |
| $\beta_2$ | - | -0.06 (-0.07--0.05) | - | - | - |
| R$^2$ | 0.40 (0.36–0.41) | 0.41 (0.36–0.42) | 0.80 (0.77–0.82) | 0.84 (0.81–0.86) | 0.93 (0.90–0.94) |
| mean of ratios[1] | 0.97 (0.96–0.97) | 0.90 (0.90–0.91) | 1.05 (1.04–1.06) | 1.01 (1.01–1.02) | 1 (1–1.01) |
| error[2] | 201% (195–223%) | 207% (201–229%) | 53% (50–55%) | 35% (33–39%) | 15% (13–18%) |

$\beta_0$ = intercept. $\beta_1$ = coefficient for E-BAM. $\beta_2$ = coefficient for temperature (T). [1]Mean of ratios of calibrated PMS3003 to E-BAM PM$_{2.5}$ conc.. [2]Defined as 1 Std.Dev of ratios of calibrated PMS3003 to E-BAM PM$_{2.5}$ conc..

**Table 3: Summary of sensor performance characteristics for the three PMS3003 PM$_{2.5}$ measurements at 1min, 1 h, 6 h, 12 h, and 24 h time intervals. The three PMS3003s were compared to the T640 sitting on the roof from June 30, 2017 to July 31, 2017 at US EPA RTP. The temperature (T) correction is only valid for the 1 min to 12 h aggregated data and the RH correction is only valid for the 1 min to 6 h aggregated data. The fit coefficients for the calibration models are provided. The R$^2$, mean of ratios, and error are performance characteristics for the calibrated PMS3003 PM$_{2.5}$ measurements after the entire suite of indicated adjustments in comparison with reference values. The results are displayed in mean (range) format. Note the mean statistics were obtained by fitting the models to the PMS3003 PM$_{2.5}$ measurements averaged across all the three sensor package units at each point in time.**

| Performance characteristics | 1 min | | | 1 h | | | 6 h | | | 12 h | | 24 h |
|---|---|---|---|---|---|---|---|---|---|---|---|---|
| adjustments | T640 | RH, T640 | RH, T640, T | T640 | RH, T640 | RH, T640, T | T640 | RH, T640 | RH, T640, T | T640 | T640, T | T640 |
| $\beta_0$ | -2.1 (-2.6--1.9) | -3.5 (-3.9--3.3) | -1.5 (-1.9--1.0) | -2.1 (-2.6--1.8) | -3.5 (-3.9--3.3) | -1.4 (-1.8--1.0) | -2.4 (-2.9--2.1) | -3.2 (-3.6--3) | -0.3 (-0.6--0.1) | -3.4 (-3.9--3) | 8.7 (8.6–8.7) | -4.1 (-4.6--3.6) |
| $\beta_1$ | 1.8 (1.7–1.9) | 1.4 (1.4–1.5) | 1.5 (1.4–1.6) | 1.8 (1.7–1.9) | 1.4 (1.4–1.5) | 1.5 (1.4–1.6) | 1.8 (1.7–2) | 1.4 (1.3–1.5) | 1.5 (1.4–1.6) | 1.9 (1.8–2.1) | 2.2 (2.1–2.4) | 2 (1.9–2.1) |
| $\beta_2$ | - | - | -0.09 (-0.1--0.07) | - | - | -0.09 (-0.1--0.08) | - | - | -0.13 (-0.14--0.11) | - | -0.49 (-0.51--0.47) | - |
| R$^2$ | 0.66 (0.63–0.67) | 0.93 (0.90–0.93) | 0.94 (0.93–0.94) | 0.66 (0.64–0.68) | 0.93 (0.92–0.94) | 0.95 | 0.73 (0.71–0.74) | 0.92 (0.91–0.93) | 0.95 (0.95–0.96) | 0.84 (0.82–0.85) | 0.93 (0.92–0.94) | 0.94 (0.93–0.94) |
| mean of ratios[1] | 0.99 | 1 | 1 | 0.99 | 1 | 1 | 1 | 1 | 1 | 1 | 1 (0.99–1) | 1 |
| error[2] | 27% (27–30%) | 11% (11–12%) | 9% (9–10%) | 27% (26–28%) | 10% (9–11%) | 8% (8–9%) | 22% (21–24%) | 10% (10–11%) | 8% (8–9%) | 15% (15–16%) | 11% (10–12%) | 9% |

$\beta_0$ = intercept. $\beta_1$ = coefficient for T640. $\beta_2$ = coefficient for temperature (T). [1]Mean of ratios of calibrated PMS3003 to E-BAM PM$_{2.5}$ conc.. [2]Defined as 1 Std.Dev of ratios of calibrated PMS3003 to E-BAM PM$_{2.5}$ conc..

Intercept and slope under the T640 adjustment define the linear relationship between the raw PMS3003 (y-axis) and T640 PM$_{2.5}$ measurements (x-axis) while under the RH and T640 adjustments define the linear relationship between the RH-adjusted PMS3003 (y-axis) and T640 PM$_{2.5}$ measurements (x-axis).





**Table 4: Summary of sensor performance characteristics for the two PMS3003 PM$_{2.5}$ measurements at 1 h, 6 h, 12 h, and 24 h time intervals during the monsoon season (Mon, June 8, 2017 to June 29, 2017), and by two different calibration methods (i.e., simple linear and quadratic equations) during the post-monsoon season (PoM, Oct 23, 2017 to Nov 16, 2017) at IIT Kanpur. The fit coefficients are provided for only the linear regression calibration models. The R$^2$, mean of ratios, and error are performance characteristics for the calibrated PMS3003 PM$_{2.5}$ measurements after the entire suite of indicated adjustments in comparison with reference values. The results are displayed in mean (range) format. Note the mean statistics were obtained by fitting the models to the PMS3003 PM$_{2.5}$ measurements averaged across all the two sensor package units at each point in time.**

| Characteristics | Method | Season | 1 h | | 6 h | | 12 h | | 24 h |
|---|---|---|---|---|---|---|---|---|---|
| adjustment | | | E-BAM | E-BAM, T | E-BAM | E-BAM, T | E-BAM | E-BAM, T | E-BAM |
| $\beta_0$ | Linear | Mon | 5.1 (3.8–6.6) | 88 (87–88) | -5.8 (-6.7--4.7) | 47 (46–49) | -6.5 (-7.4--5.5) | NA[4] | -4.5 (-4.9--3.8) |
| | | PoM | 74 (62–86) | 276 (275–277) | 65 (53–77) | 248 (246–249) | 74 (63–86) | 330 (293–366) | 82 (71–93) |
| $\beta_1$ | Linear | Mon | 1.4 (1.4–1.5) | 1.2 (1.2–1.3) | 1.7 (1.6–1.8) | 1.6 (1.5–1.6) | 1.7 (1.7–1.8) | NA[4] | 1.7 (1.6–1.7) |
| | | PoM | 1.4 (1.3–1.4) | 1.1 | 1.4 | 1.2 (1.1–1.2) | 1.3 | 1 | 1.2 (1.2–1.3) |
| $\beta_2$ | Linear | Mon | | -2.3 (-2.3--2.2) | | -1.4 (-1.5--1.4) | | NA[4] | 1.7 (1.6–1.7) |
| | | PoM | - | -7.9 (-8.4--7.4) | - | -7.0 (-7.5--6.5) | - | -10 (-12--8.1) | 1.2 (1.2–1.3) |
| R$^2$ | Linear | Mon | 0.61 | 0.61 (0.60–0.62) | 0.80 (0.79–0.81) | 0.81 (0.79–0.82) | 0.84 (0.83–0.85) | NA[4] | 0.78 (0.77–0.79) |
| | | PoM | 0.75 (0.73–0.75) | 0.78 (0.74–0.79) | 0.87 (0.84–0.87) | 0.90 (0.85–0.90) | 0.88 (0.86–0.88) | 0.89 (0.84–0.89) | 0.93 (0.89–0.93) |
| | Quadratic | PoM | 0.74 (0.71–0.74) | NA[3] | 0.86 (0.83–0.87) | NA[3] | 0.86 (0.81–0.86) | NA[3] | 0.93 (0.89–0.93) |
| mean of ratios[1] | Linear | Mon | 1.01 | 1.01 (0.97–1.01) | 1.01 | 1.01 (0.97–1.01) | 1 | NA[4] | 1 |
| | | PoM | 0.96 (0.96–0.97) | 0.99 (0.98–1.01) | 0.98 (0.97–0.98) | 0.99 (0.99–1.01) | 0.98 | 1 (0.97–1) | 0.99 |
| | Quadratic | PoM | 1 | NA[3] | 1 | NA[3] | 1 | NA[3] | 1 (0.99–1) |
| error[2] | Linear | Mon | 46% | 46% (44–46%) | 32% | 30% (29–30%) | 18% (18–19%) | NA[4] | 17% (17–18%) |
| | | PoM | 35% (33–39%) | 30% (30–34%) | 25% (23–28%) | 18% (18–22%) | 19% (18–22%) | 17% (17–20%) | 11% (11–14%) |
| | Quadratic | PoM | 24% (24–25%) | NA[3] | 16% (16–17%) | NA[3] | 12% (12–14%) | NA[3] | 9% (9–11%) |

$\beta_0$ = intercept. $\beta_1$ = coefficient for E-BAM. $\beta_2$ = coefficient for temperature (T). [1]Mean of ratios of calibrated PMS3003 to E-BAM PM$_{2.5}$ conc.. [2]Defined as 1 Std.Dev of ratios of calibrated PMS3003 to E-BAM PM$_{2.5}$ conc.. [3]No attempt was made to incorporate a temperature variable in quadratic models. [4]Temperature variable was not statistically significant at the 12 h time resolution for the monsoon data set.