# Peer review of "Field evaluation of low-cost particulate matter sensors in high and low concentration environments"

_Atmospheric Measurement Techniques, 2018_

## Referee Comment (RC1) · Anonymous Referee #1 · 8 May 2018

The paper presents a thorough evaluation of the performance of the PMS3003s PM2.5 mass sensor. It is well-written and reaches conclusions that will be useful to take in to consideration when implementing aerosol mass measurements in the field as part of air quality studies. The paper should be published after the minor concerns below are addressed.

Section 2.1: Although the dimensions of the electrical box housing the instruments is provided, it would be useful to also provide the dimensions and weight of the instruments themselves.

Page 8, line 20: change to ". . .AGREED quite well".

[Figure]

Page 8, line 26: change to "...WAS THE RH correction factor..."

Page 8, line 32: change to "...THE RH correction factor..."

Page 9, line 1: change to "...were compensated FOR by..."

Page 9, line 5: Why were the RH adjustments only made for R2 values greater than 0.4? Please add an explanation to the text.

Page 9, line 8: change to "...PARTICLE chemical composition and ..."

Page 9, lines 9 – 10: change to "...AEROSOL mass at some locations INCREASED continuously..."

Page 11, line 11: R2 is not a measured parameter. Please change the sentence.

Page 12, Lines 14 – 16: For the statement that "correlations among the five uncalibrated PMS3003 units were high...", please provide the timescale.

Page 13: lines 32 – 33: change to ..."through -3) from the Duke University site to the US EPA..."

Page 15, line 14: Figure 6 is referenced in the text before Figure 5b.

Page 16, line 11: Figure 8 is referenced in the text before Figure 7e-g.

Page 16, line 30: Omit "achieved".

Page 17, line 24: change to "...RH values MEASURED in..."

Page 18, lines 5 – 10: Is it possible to clean the sensors and see if that changes the instrument performance?

Page 18, line 25: What is meant by "...reaching the troughs of true ambient PM2.5 concentrations"?

Page 19, line 14: change to "...PARTICLE chemical,..."

---

## Referee Comment (RC2) · Anonymous Referee #2 · 11 May 2018

This paper well documents for the evaluation of low-cost PM sensors, including RH corrections, under various ambient PM concentrations. All descriptions and discussion are clear, but several points should be addressed before its publication.

Major comments:

- How the flow rate of low-cost PM sensors of this study were calibrated before, during, and after field deployments? Especially, I am wondering how well the flow rate was maintained in high PM concentrations (i.e., Kanpur)

- More specific descriptions on (1) how the instrument segregate the particles into the different size, especially for PM2.5, and (2) its efficiency and accuracy.

[Figure]

Minor & Technical comments

- Fig 2: what "Raw Sensor" means?

- Fig. 4 & other scatter plots: (1) indicate the number of used data in color scale to figure out the distribution of PM concentration, (2) add bias and root mean square difference in each figure.

- Why the slope is higher than 1 for most cases (e.g., Figs 4 and 7)? Discussion on the potential factors effecting on low-cost PM sensor measurements would be helpful in future research. - It would be nice to provide in supplement how much the total cost, including the sensor, was.

―――――――――――――――――――――

---

## Author Comment (AC1) · 14 Jun 2018

The comment was uploaded in the form of a supplement:
https://www.atmos-meas-tech-discuss.net/amt-2018-111/amt-2018-111-AC1-supplement.pdf

---

## Author Comment (AC2) · 14 Jun 2018

The comment was uploaded in the form of a supplement:
https://www.atmos-meas-tech-discuss.net/amt-2018-111/amt-2018-111-AC2-supplement.pdf

---

## Author Response (AR1)

**Response to Comments from Reviewer #1 AMT-2018-111**

The authors would like to first and foremost thank the reviewer #1 for the careful perusal of the manuscript and the insightful comments which helped improve the manuscript. The reviewer's comments are in italics, the summaries of our responses are in plain font, and the changes in the manuscript are in red text. Page and line numbers refer to the original document.

5 **Reviewer #1**

*Section 2.1: Although the dimensions of the electrical box housing the instruments is provided, it would be useful to also provide the dimensions and weight of the instruments themselves.*

**Response:** Thank you for pointing this out, we agree that the weights of the enclosure and the instrument itself are important as the weight is one of the major design criteria that low-cost air quality packages are required to meet, and the lightweight

10 characteristic gives emerging low-cost devices an advantage over traditional instrumentation. We have added the weight information about both the enclosure and the instrument itself (the PMS3003 PM sensor) in Section 2.1 along with the dimension of the instrument.

Modified text in Section 2.1 (additions and changes in bold):

"The Plantower PMS3003 sensor **(dimension: 5.0 cm L × 4.3 cm W × 2.1 cm H; weight: 40 g)** along with a Sparkfun

15 SHT15 RH and temperature sensor, a Teensy 3.2 USB-based microcontroller, a ChronoDot V2.1 high precision real-time clock, a microSD card adapter, a Pololu 5V S7V7F5 voltage regulator, a DC barrel jack connector, and a basic 5 mm LED was connected to a custom designed printed circuit board (PCB), shown in Fig. 1a. We programmed the Teensy 3.2 microcontroller to measure PM mass concentrations ($\mu$g m$^{-3}$) every second and to store the time-stamped 1 min averaged measurements to text files on a microSD card. To protect sensors from rain and direct sunlight, all components were housed

20 in a 20.50 cm L × 9.95 cm W × 6.70 cm H**, 363 g lightweight** NEMA (National Electrical Manufacturers Association) electrical box (Bud Industries NBF32306) as shown in Fig. 1b. The inlet of the Plantower sensor was aligned with a hole drilled in the electrical box to ensure unrestricted airflow into the sensor. **Each Duke PM air quality monitoring package is estimated to weigh ~430 g in total and was** continuously powered up by **a** 5V 1A USB wall **charger**. The total material costs for one PM monitoring package including the Plantower PMS3003 sensor, the supporting circuitry, the enclosure, and

25 additional power cords are approximately USD 200. More detailed instructions on how to assemble the sensor packages and information on how to use their data can be found on our webpage (http://dukearc.com)."

*Page 8, line 20: change to ". . .AGREED quite well".*

**Response:** Thank you, we have made the suggested change to wording.

Modified text in Page 8, line 20 (changes in bold):

"They found that the model **agreed quite well with the** field data **both** collected from their study and **from** a previous study (Day and Malm, 2000)."

*Page 8, line 26: change to "...WAS THE RH correction factor..."*

**Response:** Thank you, we have corrected the grammatical error.

Modified text in Page 8, line 26 (change in bold):

"Ordinary least squares (OLS) regressions were conducted to obtain the empirical regression parameters $a$ and $b$ in Eq. (1), where the dependent variable was **the** RH correction factors calculated as the ratio of…"

*Page 8, line 32: change to "...THE RH correction factor..."*

**Response:** Thank you, we have corrected the grammatical error.

Modified text in Page 8, line 32 (change in bold):

"The empirical equations derived were used to compute **the** RH correction factor for a given RH…"

*Page 9, line 1: change to "...were compensated FOR by..."*

**Response:** Thank you, we have corrected the grammatical error.

Modified text in Page 9, line 1 (change in bold):

"The RH interferences were compensated **for** by dividing…"

*Page 9, line 5: Why were the RH adjustments only made for R2 values greater than 0.4? Please add an explanation to the text.*

**Response:** A relatively high correlation value (i.e., $R^2 = 0.4$) of the empirical equation for computing the RH correction factors (i.e., Eq. (1)) was chosen as the cut-off point in this study because we want to ensure that the RH corrections can indeed lower the error of the low-cost sensor $PM_{2.5}$ measurements. Given the poor precision of the E-BAM, lower correlation values may lead to marginal (if any) improvements in the accuracy of the low-cost sensor measurements. This theory can be corroborated by the temperature correction results from the current study: An AIC difference of 2 is a standard threshold for model selection. However, even when the AIC indicated that the temperature predictor was statistically significant in the calibration model, the temperature correction still resulted in marginal (Kanpur monsoon 6 h results, see Table 4) or no (Kanpur monsoon 1 h results, see Table 4) or negative (Duke 1 h results, see Table 2) improvements. Furthermore, the highest correlation of the empirical RH correction factor equation obtained at sites using an E-BAM as the reference monitor was 0.13 (Kanpur monsoon 1 h results, see Fig. S7). This value is too low to warrant conducting the RH correction (even if we lowered the cut-off point to a non-ideal 0.20). Other users who have access to more precise regulatory-grade instruments can choose to lower this threshold as they see fit. We agree that the original text lacks the corresponding justification and is therefore unclear. We have added an explanation to the original text.

Modified text in the 3rd paragraph of Section 2.3.1 (additions in bold):

"We only performed the RH adjustments when the fitted models for any of the sampling locations over any time averaging interval had at least a moderate coefficient of determination ($R^2 \geq 0.40$). **The slightly high correlation cut-off value was implemented in this study to ensure that the RH corrections can effectively lower the error of the low-cost sensor**

**PM$_{2.5}$ measurements.** Despite the similarity of the general shape of correction factor curves in different studies, the detailed behaviors of aerosols diverged greatly due to considerable difference in particles' chemical composition and diameter (Waggoner et al., 1981; Zhang et al., 1994; Day and Malm, 2000; Chakrabarti et al., 2004; Soneja et al., 2014). In a previous study (Day and Malm, 2000), aerosols mass at some locations began to increase continuously above a relatively low RH (such as 20%), whereas at other locations it exhibited a distinct deliquescent behavior (i.e., aerosols water uptake occurred at a relatively high RH). Even for aerosols showing deliquescent behavior, the observed deliquescence RH (RH threshold) varies from study to study. Soneja et al. (2014) also found underestimation of PM concentrations (correction factors less than 1) below 40% RH. Because of these uncertainties, we conducted RH adjustments across the entire range of recorded RH without incorporating an RH threshold. Additionally, the RH adjustments in this study were always performed separately from and prior to either temperature adjustments or reference monitor adjustments."

*Page 9, line 8: change to ". . .PARTICLE chemical composition and . . ."*

**Response:** Thank you, we have made the suggested change to wording.

Modified text in Page 9, line 8 (change in bold):

"…the detailed behaviors of aerosols diverged greatly due to considerable difference in **particle** chemical composition and diameter…"

*Page 9, lines 9 – 10: change to ". . .AEROSOL mass at some locations INCREASED continuously. . ."*

**Response:** Thank you, we have made the suggested change to wording.

Modified text in Page 9, lines 9–10 (changes in bold):

"…**aerosol** mass at some locations **increased** continuously above a relatively low RH…"

*Page 11, line 11: R2 is not a measured parameter. Please change the sentence.*

**Response:** Thank you for pointing this out. We agree that all the performance metrics including $R^2$, RMSE, MAE, and MBE are calculated rather than measured parameters. We have revised the language.

Modified text in Page 11, line 11 (change in bold):

"To date, only a few studies have attempted to **compute** parameters other than $R^2$ to gauge the overall performance of low-cost sensor technologies."

*Page 12, Lines 14 – 16: For the statement that "correlations among the five uncalibrated PMS3003 units were high. . .", please provide the timescale.*

**Response:** Thank you, the timescale has been added.

Modified text in Page 12, lines 14–16 (additions in bold):

"Correlations among the five uncalibrated PMS3003 units were high ($R^2$ = 0.98–1.00) **on 1 h timescale** even under low ambient PM$_{2.5}$ concentrations with slopes averaging 1 ± 0.1 and negligible intercepts averaging 0.3 ± 0.3 (Fig. S1), suggesting excellent intra-PMS3003 precision."

*Page 13: lines 32 – 33: change to . . ."through -3) from the Duke University site to the US EPA. . ."*

**Response:** Thank you, we have made the suggested change.

Modified text in Page 13, lines 32–33 (changes in bold):

"…we moved three PMS3003 units (labeled PMS3003-1 through -3**)** from the Duke University study site-to the US EPA…"

*Page 15, line 14: Figure 6 is referenced in the text before Figure 5b.*

**Response:** Thank you for pointing this out. We have revised Section 3.2.1 and 3.2.2 by moving the descriptions of both
Figure 5a and 5b to the beginning of Section 3.2.1, before the first reference to Figure 6 (at the beginning of Section 3.2.2).
Modified text in Section 3.2.1 and 3.2.2 (additions and changes in bold):

**"3.2.1 PM$_{2.5}$ concentration, RH, and temperature on 1 h scale**

[revised manuscript text omitted]

*Page 18, lines 5 – 10: Is it possible to clean the sensors and see if that changes the instrument performance?*

**Response:** Thank you for suggesting this possibility. Unfortunately, we have not attempted to clean the sensors throughout the current Kanpur field test. We acknowledge that the effect of PM deposition on the low-cost PM sensor performance and calibration particularly in areas of high ambient PM concentrations (e.g., Kanpur) is understudied. Considering the substantial implications of this research topic for the development and maintenance of future low-cost PM sensors networks in environments such as polluted urban areas, we believe a separate, specialized, and well-designed field campaign is required for a rigorous evaluation. Also given the present long length of the manuscript, we also inclined not to expand on this complicated issue. However, we are planning to address this issue by determining if routine cleaning (e.g., gently blowing through the low-cost sensor with canned air) will be helpful for maintaining or improving the sensor performance in a forthcoming publication. We have added additional text to the 1st paragraph of Section 3.3.2 to clarify our points.

Modified text in Section 3.3.2 (additions in bold):

"As with the two field tests in the low concentration region, the two PMS3003 units were highly correlated with each other during both the monsoon ($R^2 = 0.99$) and post-monsoon seasons ($R^2 = 0.93$) in Kanpur (Fig. S6). This good agreement is also reflected in Fig. 9, which displays that the two sensors were in sync and tracked reasonably well with the E-BAM. However, there was a minor decrease in the intra-sensor correlation from the monsoon to post-monsoon seasons that might signal a performance change of the two PMS3003 sensors either due to minor deterioration or a change in the pollutant source. Figure S6 illustrates that the magnitude of the deviation from the regression line during the monsoon season was likely irrelevant to the deployment time (measured by the number of hours past the beginning of the Kanpur study, i.e., 2017 June 08 00:00). In contrast, the extent of the divergence was somewhat larger for the longer deployment time near the high end of the $PM_{2.5}$ range over the post-monsoon period. One plausible explanation for the distinguishable post-monsoon (but not monsoon season) change is the routine exposure (for nearly a month) of the sensors to high concentrations of accumulation mode aerosols. This may be especially detrimental to PM sensors; all the more so because the foggy condition during post-monsoon and winter over Kanpur may further exacerbate the accumulation of aerosol particles at lower surfaces and therefore the deposition of particles within the sensors (Li et al., 2015; Bran and Srivastava, 2017). This constant exposure

possibly caused disproportionately large detection errors primarily near the upper end of the PM$_{2.5}$ range. **The effect of PM deposition on the low-cost PM sensor performance and calibration particularly in areas of high ambient PM concentrations (e.g., Kanpur) was not evaluated as part of this work. Future studies will present how preventive maintenance of low-cost sensors including periodic cleaning can benefit their performance.** Another possible

5   explanation is the change of dominant pollutant source from the early stage of monsoon (long-range transport of mineral dust from Iran, Afghanistan, Pakistan, and the Thar Desert) to post-monsoon (local impact of biomass burning emissions) season (Ram et al., 2010). Sensors are likely to respond differently to different varieties of aerosols and the change in sensor responses might be most pronounced near the upper end of the PM$_{2.5}$ range. Figure 9b substantiates the potential change by showing that the two uncalibrated PMS3003s were unable to match the troughs of the E-BAM (even troughs below 40 μg m$^{-}$

10   $^{3}$) throughout the post-monsoon season, as they were during the monsoon season in Fig. 9a."

*Page 18, line 25: What is meant by ". . .reaching the troughs of true ambient PM2.5 concentrations"?*

**Response:** The troughs mean the local minima of the true ambient PM2.5 concentrations. We have changed the terminology from troughs to local minima throughout the manuscript for clarity.

Modified text in Page 18, lines 14–16 (changes in bold):

15   "Figure 9b substantiates the potential change by showing that the two uncalibrated PMS3003s were unable to match the **local minima** of the E-BAM (even **local minima** below 40 μg m$^{-3}$) throughout the post-monsoon season, as they were during the monsoon season in Fig. 9a."

Modified text in Page 18, lines 24–25 (change in bold):

"This appreciable drift in baseline agreed with the sensors being incapable of reaching the **local minima** of true ambient

20   PM$_{2.5}$ concentrations."

Modified text in Page 20, lines 7–9 (change in bold):

"Figure S8 shows that the quadratic model might suit the post-monsoon 1 h aggregated data better than the simple linear model as the simple linear model failed to capture the **local minima** of the E-BAM throughout the post-monsoon period."

*Page 19, line 14: change to ". . .PARTICLE chemical,. . ."*

25   **Response:** Thank you, we have made the suggested change to wording.

Modified text in Page 19, line 14 (change in bold):

"…great reliance on **particle** chemical, microphysical, and optical properties (Laulainen, 1993)."

**Response to Comments from Reviewer #2 AMT-2018-111**

The authors would like to first and foremost thank the reviewer #2 for the careful perusal of the manuscript and the insightful comments which helped improve the manuscript. The reviewer's comments are in italics, the summaries of our responses are in plain font, and the changes in the manuscript are in red text. Page and line numbers refer to the original document.

5 **Reviewer #2**

*- How the flow rate of low-cost PM sensors of this study were calibrated before, during, and after field deployments? Especially, I am wondering how well the flow rate was maintained in high PM concentrations (i.e., Kanpur)*

**Response:** Thank you for bringing up this question. The configuration of the PMS3003 sensors suggests that their detection approach is volume scattering of the particles rather than light scattering at the single particle level. This volume scattering

10 detection approach determines that the PMS3003 sensors' PM measurements are roughly independent of flow rate. We have added two sentences in the 1$^{st}$ paragraph of Section 2.1 to clarify this issue.

Modified text in the 1$^{st}$ paragraph of Section 2.1 (additions in bold):

"The low-cost sensors evaluated in the present study are Plantower particulate matter sensors (model PMS3003). The Plantower PMS3003 sensors were chosen because 1) they are priced at a small fraction of the cost of reference monitors

15 (approximately USD 30) and 2) their manufacturer reported maximum errors are relatively low ($\pm 10$ $\mu$g m$^{-3}$ in the 0–100 $\mu$g m$^{-3}$ range, and $\pm 10\%$ in the 100–500 $\mu$g m$^{-3}$ range). The sensors employ a light-scattering approach to measure PM$_1$, PM$_{2.5}$, and PM$_{10}$ mass concentrations in real-time. Ambient air laden with different-sized particles is drawn into the sensor measurement volume where the particles are illuminated with a laser beam, and the resulting scattered light is measured perpendicularly by a recipient photo-diode detector. These raw light signals are filtered and amplified via electronic filters

20 and circuitry before being converted to mass concentrations. The manufacturer datasheet indicates that the measurement range of this specific sensor model spans from 0.3 $\mu$m to 10 $\mu$m. **The configuration of the PMS3003 sensors suggests that their detection approach is volume scattering of the particle population rather than light scattering at the single particle level. This volume scattering detection approach results in PM measurements that are independent of flow rate.** PM mass concentration measurements either with or without a manufacturer "atmospheric" calibration are available

25 from the Plantower sensor outputs. Nevertheless, the manufacturer did not provide any documentation to elaborate on how the calibration algorithm was derived. The influence of meteorological factors (e.g., RH, temperature) was likely not accounted for in the manufacturer calibrations. Therefore, we used the sensor reported PM concentration estimates without an "atmospheric" calibration in the current study. Prior to field deployment, no attempt was made to calibrate these sensors under laboratory conditions due to a potentially marked discrepancy in particle size, composition, and optical properties of

30 field and laboratory conditions."

*- More specific descriptions on (1) how the instrument segregate the particles into the different size, especially for PM2.5, and (2) its efficiency and accuracy.*

**Response:** Thank you for raising this question. We apologize for not mentioning in the text that the Plantower PMS3003 PM sensors (unlike their PMS1003 and PMS5003 counterparts) are not designed as single particle counters (i.e., the Plantower PMS3003 PM sensors report only PM mass concentration but not count). We believe the allocation of light scattering to $PM_1$, $PM_{2.5}$, and $PM_{10}$ is based on Plantower's proprietary algorithm, which the manufacturer does not disclose to the public. Therefore, comparing the PMS3003's particle size distribution with that of regulatory-grade instrument (e.g., GRIMM) is impractical and beyond the scope of the current manuscript. Since PMS's particle size distribution is obtained based on a theoretical model rather than an actual measurement (a physical segregation), we would tend to conclude that the efficiency of the segregation is not applicable to the PMS series devices. The primary focus of the current manuscript is how accurately the PMS3003 PM sensors can measure $PM_{2.5}$ mass concentration (rather than particle count) after calibration by co-location with research-grade instruments and after adjustment to meteorology interferences (if available). The evaluation of mass concentration instead of particle count takes priority because only the PM mass concentration (but not particle count) is regulated and monitored under the National Ambient Air Quality Standards (NAAQS) at this stage. We regret that we are unable to comment on the accuracy of PMS3003's theoretical segregation. However, Kelly et al. (2017) thoroughly compared the size distribution provided by the PMS1003 sensors with those provided by a GRIMM Model 1.109 (a portable aerosol spectrometer). The results can be considered as a rough representative estimate of the segregation accuracy of the PMS series devices. In particular, Kelly et al. (2017) demonstrated that "the PMS1003 sensors overestimate total daily average PM counts by a factor of 1.5–2.4 compared to the GRIMM" and "the PMS overestimates particle counts by a factor of (1.1–1.9) for the 0.3 μm bin and increasingly overestimates particle counts as particle size increases". We have added two sentences in the 1$^{st}$ paragraph of Section 2.1 to be more upfront about the characteristics of the PMS3003 sensors.

Modified text in the 1$^{st}$ paragraph of Section 2.1 (additions in bold):

"The low-cost sensors evaluated in the present study are Plantower particulate matter sensors (model PMS3003). The Plantower PMS3003 sensors were chosen because 1) they are priced at a small fraction of the cost of reference monitors (approximately USD 30) and 2) their manufacturer reported maximum errors are relatively low ($\pm10$ μg m$^{-3}$ in the 0–100 μg m$^{-3}$ range, and $\pm10\%$ in the 100–500 μg m$^{-3}$ range). **Unlike their PMS1003 and PMS5003 counterparts, the PMS3003s are not designed as single particle counters.** The sensors employ a light-scattering approach to measure $PM_1$, $PM_{2.5}$, and $PM_{10}$ mass concentrations in real-time **and are believed to apportion light scattering to $PM_1$, $PM_{2.5}$, and $PM_{10}$ based on a confidential proprietary algorithm (Kelly et al., 2017)**. Ambient air laden with different-sized particles is drawn into the sensor measurement volume where the particles are illuminated with a laser beam, and the resulting scattered light is measured perpendicularly by a recipient photo-diode detector. These raw light signals are filtered and amplified via electronic filters and circuitry before being converted to mass concentrations. The manufacturer datasheet indicates that the measurement range of this specific sensor model spans from 0.3 μm to 10 μm. PM mass concentration measurements either

with or without a manufacturer "atmospheric" calibration are available from the Plantower sensor outputs. Nevertheless, the manufacturer did not provide any documentation to elaborate on how the calibration algorithm was derived. The influence of meteorological factors (e.g., RH, temperature) was likely not accounted for in the manufacturer calibrations. Therefore, we used the sensor reported PM concentration estimates without an "atmospheric" calibration in the current study. Prior to field deployment, no attempt was made to calibrate these sensors under laboratory conditions due to a potentially marked discrepancy in particle size, composition, and optical properties of field and laboratory conditions."

*- Fig 2: what "Raw Sensor" means?*

**Response:** In Fig. 2, "raw sensor PM$_{2.5}$ measurements" means uncalibrated sensor PM$_{2.5}$ measurements. We have added one sentence within the caption of Fig. 2 on page 29 to clarify this.

Modified text within the caption of Fig. 2 on page 29 (additions in bold):

"Figure 1: Flow path for sensor calibrations. **Note raw sensor PM$_{2.5}$ measurements are uncalibrated sensor PM$_{2.5}$ measurements.**"

*- Fig. 4 & other scatter plots: (1) indicate the number of used data in color scale to figure out the distribution of PM concentration, (2) add bias and root mean square difference in each figure.*

**Response:** Thank you for suggesting these changes to the scatter plots in the current manuscript.

1) We have shown the distribution of PM$_{2.5}$ concentration in all the scatter plots by adding marginal rugs on both x- and y-axis. Within all the figure captions for the scatter plots, we have also described explicitly that marginal rugs were added to better visualize the distribution of data on each axis. Thus, we feel that these scatter plots do not warrant any additional visualization to illustrate the distribution of PM concentration.

2) We thank the reviewer for suggesting these two performance metrics for low-cost sensor evaluation. First of all, mean bias error (MBE) defined as $\frac{1}{n}\sum_{i=1}^{n}(\hat{y}_i - y_i)$, where $\hat{y}_i$ is the calibrated PMS3003 PM$_{2.5}$ mass concentrations and $y_i$ is the reference monitor PM$_{2.5}$ mass concentrations, is equivalent to the mean of ratios presented in the current paper (defined as the average of ratios of the calibrated PMS3003 PM$_{2.5}$ mass concentrations to reference monitor values). Although not obvious, calibrated PMS3003 PM$_{2.5}$ values always have an MBE of 0 (i.e., $\hat{y}_{i_{mean}} = y_{i_{mean}}$) using a simple linear regression calibration equation and an MBE roughly close to 0 (i.e., $\hat{y}_{i_{mean}} \cong y_{i_{mean}}$) using a quadratic calibration equation. This is equivalent to say calibrated PMS3003 PM$_{2.5}$ values should always have a mean of ratios close to 1 (i.e., $\hat{y}_{i_{mean}} \cong y_{i_{mean}}$). The only difference between MBE and mean of ratios is that the former one is expressed in a subtraction form while the latter one in a division form. Second, we agree that RMSE is a standard and widely used performance score. Yet, unlike percent error used in the current manuscript, RMSE is generally unfavourable for comparison across different data sets (particularly data sets from different field campaigns with drastically different PM concentrations). Specifically, calibrated PMS3003 PM$_{2.5}$ values at a 1 h time resolution had an average RMSE of 10.4±0.2 μg m$^{-3}$, 12.7±0.1 μg m$^{-3}$, and 31.0±0.9 μg m$^{-3}$ at the Duke site, IIT Kanpur site during monsoon, and post-monsoon season, respectively. However, given that the hourly averaged ambient PM$_{2.5}$ levels at IIT Kanpur during post-monsoon (116±57 μg m$^{-3}$) were roughly 13 times

higher than that at the Duke site ($9\pm9$ µg m$^{-3}$), it is unfair to conclude that the sensors had the best performance at the Duke site while the worst performance at the IIT Kanpur site during post-monsoon based on RMSEs. We tend to believe that a normalized metrics such as the percent error in this study is more straightforward and more suited for observing the trend in measurement accuracy with PM concentrations. We tend to hold that RMSE is more appropriate to be used in model

5    selection given a same data set as described in Section 3.3.3 and 3.3.4 (see Table S2 and S3). Overall, mean of ratios defined in this study are equivalent to MBE and the error defined in this study is a legitimate quantitative performance criterion to more intuitively validate that PMS3003 sensors can have credible data and good accuracy after proper calibrations based on reference monitors and corrections for meteorological interferences. Therefore, we feel that these scatter plots do not warrant additional performance metrics such as bias and RMSE. However, during our calculation of the RMSEs prompted by this

10    question, we found that we have incorrectly defined RMSE as the standard deviation of differences between calibrated and raw PMS3003 PM$_{2.5}$ mass concentrations (it should be the difference between calibrated PMS3003 and reference monitor PM$_{2.5}$ mass concentrations) in Section 2.3.3 on page 11 in the original manuscript and consequently incorrectly calculated RMSEs for Table S2 and S3 on page 10 in the original supplementary document. This is a major oversight on our side and we deeply regret for this mistake and have made the corresponding corrections. In particular, the correct recalculation has

15    significantly lowered the RMSEs by up to 15 µg m$^{-3}$ for 1 h quadratic method in Table S2 and for 1 h linear method in Table S3. The revision did not change the conclusion of the manuscript.

Modified text in Section 2.3.3 on page 11, lines 2–3 (changes in bold):

"where n is the number of observations, $\hat{y}_i$ is the calibrated PMS3003 PM$_{2.5}$ mass concentrations, and $y_i$ is the **reference monitor** PM$_{2.5}$ mass concentrations."

20    Modified text in Section 3.3.3 on page 20, lines 16–19 (changes in bold):

"Even when the nonlinearity was not strong enough to make the simple linear fit statistically different from the quadratic fit (i.e., the quadratic term $a_2$ in the quadratic fit (Eq. (7)) not significantly different from 0 with p>0.1) at 24 h integration time, the quadratic fit can still reduce the mean error and **the range of RMSEs** by 2% (Table 4)**, and 3** µg m$^{-3}$ (Table S2), respectively."

25    Modified Table S2 in supplementary document on page 10 (changes in bold):

Table S1: Summary of AIC and RMSE (goodness of fit and accuracy estimates) for the two E-BAM calibrated PMS3003 PM$_{2.5}$ responses during the post-monsoon season at IIT Kanpur, using the simple linear and quadratic calibration methods as a function of time averaging intervals. The results are displayed in mean (range) format. Note the mean statistics were obtained by fitting the models to the PMS3003

30    PM$_{2.5}$ measurements averaged across the two sensor package units at each point in time. The model that had the best goodness of fit and accuracy estimates at each averaging time interval is indicated with shading.

| Timescales | 1 h | | 6 h | | 12 h | | 24 h | |
|---|---|---|---|---|---|---|---|---|
| Method | Linear | Quadratic | Linear | Quadratic | Linear | Quadratic | Linear | Quadratic[1] |
| AIC | 5731 | 5670 | 932 | 916 | 462 | 454 | 214 | 210 |
| | (5731–5778) | (5670–5720) | (932–949) | (916–933) | (462–474) | (454–469) | (214–229) | (210–225) |
| RMSE | **31** | **27** | **21** | **19** | **19** | **18** | **13** | **13** |
| | **(31–33)** | **(27–28)** | **(21–23)** | **(19–21)** | **(19–21)** | **(18–21)** | **(13–17)** | **(13–14)** |

[1]The quadratic term ($a_2$) in the quadratic fit (Eq. (7)) for the PMS3003-6 was not significantly different from 0 ($p>0.1$).

Modified Table S3 in supplementary document on page 10 (changes in bold):

Table S2: Summary of AIC and RMSE (goodness of fit and accuracy estimates) for the E-BAM calibrated PMS3003 $PM_{2.5}$ results of the pooled Duke University and IIT Kanpur data sets, using the simple linear and quadratic calibration methods as a function of time averaging intervals. Note the values are statistics for the averaged sensor models, which were obtained by fitting the models to the means of PMS3003 $PM_{2.5}$ measurements averaged across all sensor package units during each sampling period. The model that had the best goodness of fit and accuracy estimates at each averaging time interval is indicated with shading.

| Timescales | 1 h | | 6 h | | 12 h | | 24 h | |
|---|---|---|---|---|---|---|---|---|
| Method | Linear | Quadratic | Linear | Quadratic | Linear | Quadratic | Linear | Quadratic |
| AIC | 21005 | 20638 | 3376 | 3225 | 1697 | 1590 | 836 | 764 |
| RMSE | 18 | 17 | 12 | 11 | 12 | 10 | 10 | 8 |

[*]All the models' coefficients were statistically significant ($p<0.1$).

- *Why the slope is higher than 1 for most cases (e.g., Figs 4 and 7)? Discussion on the potential factors effecting on low-cost PM sensor measurements would be helpful in future research. - It would be nice to provide in supplement how much the total cost, including the sensor, was.*

**Response:**

1) Thank you for bringing up this question. The slope is higher than 1 for most cases suggests that Plantower PMS3003s mostly overestimate ambient $PM_{2.5}$ mass concentrations prior to calibration based on reference monitors. We regret that we are unable to comment on the underlying reasons for this overestimation since Plantower PMS3003s allocate light scattering to $PM_1$, $PM_{2.5}$, and $PM_{10}$ mass concentrations based on Plantower's confidential proprietary algorithm (as described in our response to your second comment). Plantower is not clear on whether, and how the sensors may be calibrated prior to shipping them to customers. We would like to emphasize that the important question is how well the $PM_{2.5}$ mass concentrations made by these low-cost sensors after calibration by collocation and adjustments to meteorological interferences can compare to the true ambient $PM_{2.5}$ values reported by reference monitors. And we have validated in the current manuscript that these PMS3003 sensors can have credible $PM_{2.5}$ data and good accuracy after proper calibrations based on ideal reference monitors and corrections for meteorological biases.

2) Thank you for the suggestion. We have added a sentence in Conclusion on page 22, line 1 to make our discussion more prominent.

Modified text in Conclusion on page 22, line 1 (additions in bold):

"Even though the RH correction factor models might be highly location-specific, it is striking to see that they were capable of explaining up to nearly 30% of the variance in 1 min, 1 h and 6 h aggregated sensor measurements and reducing mean errors down from ~22–27% to roughly 10% even at the finest 1 min time resolution. Compared to the RH corrections, temperature corrections were found to be relatively small and can only scale uncertainties down by 7% at most; however, in addition to the other corrections this may help to achieve the highest possible accuracy level. It is important to note that the success of both RH and temperature corrections relies on the precision of reference instruments. **Properly accounting for these systematic meteorology-induced influences is helpful in making high quality $PM_{2.5}$ measurements at a low cost.**"

3) Thank you for the suggestion. The cost of Plantower PMS3003 sensor itself has already been mentioned in Section 2.1 on page 4, lines 22–23, which reads "…1) they are priced at a small fraction of the cost of reference monitors (approximately USD 30) …". The total cost of a full Duke PM air quality monitoring package has also been mentioned in Section 2.1 on page 5, lines 14–16, which reads "The total material costs for one PM monitoring package including the Plantower PMS3003 sensor, the supporting circuitry, the enclosure, and additional power cords are approximately USD 200.". Prompted by this comment, we have added the costs of the Plantower PMS3003 sensor, the supporting circuitry, the enclosure, and the power cords to this sentence in Section 2.1 on page 5, lines 14–16 to clarify the breakdown of the total costs.

Modified text in Section 2.1 on page 5, lines 14–16 (additions in bold):

[revised manuscript text omitted]

---

## Author Response (AR2)

**Response to Comments from Associate Editor AMT-2018-111**

The authors would like to thank the associate editor for the careful review of the revised manuscript and the valuable comments which helped improve the manuscript. The editor's comments are in italics, the summaries of our responses are in plain font, and the changes in the manuscript are in red text. Page and line numbers refer to the revised document.

**5 Associate Editor**

- Considering the AMT discussion and to finalize the manuscript, I would simply ask whether you would consider revising the last sentences in your conclusions in order to make a stronger statement on the need for calibration for these low-cost sensors. For example, should the sentence "Overall, we conclude that the Plantower PMS3003 sensors, as a promising lowcost PM monitor, can achieve high accuracy and precision over a wide range in PM2.5 concentration, but only after

- 10 applying appropriate calibration" be turned into "Appropriate calibration is required to achieve the required level of accuracy and precision over a wide range in PM2.5 concentration needed for monitoring PM2,5 with Plantower PMS3003 sensors". Same for the next sentence in some ways : "is ONLY feasible with current low-cost sensing technology if proper calibrations is performed..." It basically says the same but I am afraid that considerations on the need for adequate traceability of calibration / RH adjustments will be lost somewhere in the process and that your paper may be only
- 15 considered as a validation of sensor's performances for PM monitoring. Conditions for reaching the expected performances should not be seen as optional, especially when large cities in emerging countries are considering the use of such sensors for their air quality operational framework.

**Response:** Thank you for the insightful and constructive suggestion. We agree that the adequate calibration and meteorological parameter adjustments should not be seen as optional. This message is critical to anyone who considers using

20 such sensors for various applications and should be made more prominent. Therefore, as suggested, we believe the overall tone of the conclusion should be stronger. We have modified the entire 3rd paragraph of Conclusion to accommodate this issue.

Modified text in the 3rd paragraph of Conclusion on pages 22–23 (additions and changes in bold):

"Overall, we conclude that appropriate calibration models using ideal reference monitors and dynamic adjustments for meteorological parameters are an essential prerequisite for the Plantower PMS3003 sensors to achieve high accuracy and precision over a wide range in PM2.5 concentration typically encountered in the ambient monitoring. After proper calibration, the Plantower PMS3003 low-cost PM sensors are promising monitors for dense, wireless, real-time PM sensor network development in hazy urban areas such as Delhi and Mumbai, India to complement the existing networks by better approximating the location of major PM2.5 sources (local vs. regional) and by advancing our **understanding of** the influence of meteorology such as specific wind patterns on the resulting regional PM2.5 levels in order to guide local and regional air quality management (Hagler et al., 2006)."

**Field evaluation of low-cost particulate matter sensors in high and low concentration environments**

Tongshu Zheng1, Michael H. Bergin1, Karoline K. Johnson1, Sachchida N. Tripathi2, Shilpa Shirodkar2, Matthew S. Landis3, Ronak Sutaria4, David E. Carlson1,5

[revised manuscript text omitted]
_{2.5}] = 36 \pm 17\mu g \text{ m}^{-3}$ ; RH =  $62 \pm 15\%$ ; temperature =  $33 \pm 5^{\circ}\text{C}$ ) and post-monsoon seasons ( $[PM_{2.5}] = 116 \pm 57\mu g \text{ m}^{-3}$ ; RH =  $63 \pm 16\%$ ; temperature =  $22 \pm 4^{\circ}\text{C}$ ) and two suburban settings in Durham ( $[PM_{2.5}] = 9 \pm 16\%$ )

30 ([PM2.5] =  $116 \pm 57\mu \text{g m}^{-3}$ ; RH =  $63 \pm 16\%$ ; temperature =  $22 \pm 4^{\circ}\text{C}$ ) and two suburban settings in Durham ([PM2.5] =  $9 \pm 9\mu \text{g m}^{-3}$ ; RH =  $45 \pm 19\%$ ; temperature =  $15 \pm 8^{\circ}\text{C}$ ) and RTP, NC, US ([PM2.5] =  $10 \pm 3\mu \text{g m}^{-3}$ ; RH =  $64 \pm 22\%$ ; temperature

[revised manuscript text omitted]